# Single-cell transcriptomic analysis identifies murine heart molecular features at embryonic and neonatal stages

Wei Feng [1,3], Abha Bais[1,3], Haoting He[1], Cassandra Rios [1], Shan Jiang[1], Juan Xu[1], Cindy Chang[1], Dennis Kostka [1,2] & Guang Li [1]✉

Heart development is a continuous process involving significant remodeling during embryogenesis and neonatal stages. To date, several groups have used single-cell sequencing to characterize the heart transcriptomes but failed to capture the progression of heart development at most stages. This has left gaps in understanding the contribution of each cell type across cardiac development. Here, we report the transcriptional profile of the murine heart from early embryogenesis to late neonatal stages. Through further analysis of this dataset, we identify several transcriptional features. We identify gene expression modules enriched at early embryonic and neonatal stages; multiple cell types in the left and right atriums are transcriptionally distinct at neonatal stages; many congenital heart defect-associated genes have cell type-specific expression; stage-unique ligand-receptor interactions are mostly between epicardial cells and other cell types at neonatal stages; and mutants of epicardium-expressed genes *Wt1* and *Tbx18* have different heart defects. Assessment of this dataset serves as an invaluable source of information for studies of heart development.

The mammalian heart is the first organ to develop during embryogenesis. When it is fully developed, it consists of four chambers: left atrium (LA), right atrium (RA), left ventricle (LV), and right ventricle (RV). Each chamber is composed of three layers of tissue named the epicardium, myocardium, and endocardium. The myocardium is made up of several cell types, including cardiomyocytes (CMs), fibroblasts (Fbs), and coronary vasculature-related cell types such as vascular endothelial cells (Vas_ECs), smooth muscle cells (SMCs), and pericytes[1-3]. The epicardium consists mainly of epicardial cells (Epis), and the endocardium contains endocardial endothelial cells (Endo_ECs). Additionally, the heart contains several other cell types, such as valve cells and conduction cells[1-4].

The specification of each cell type in mice follows their own developmental trajectories. Early in embryogenesis, CMs develop from both heart progenitor fields, the first and second heart fields, unequally contributing to the CM lineage development in each chamber[3,5,6]. Epis arise from proepicardium and Endo_ECs develop from precardiac mesoderm while Fbs form from both[7-9]. Vas_ECs develop from two main sources: Endo_ECs and the sinus venosus[10-12]. SMCs are reported to develop from pericytes and Epis[13,14]. The CMs and SMCs were also reported to develop from neural crest cells[15]. Important non-chamber structures like cardiac valves derive from endocardial cushions located in the atrioventricular canal and outflow tract early in development resulting in the valve primordia by E11.5. By E16.5–17.5, it further develops into valve leaflets consisting of Endo_ECs and valve interstitial cells (VICs)[16-18]. Similar to valve cells, other cell lineages continuously remodel their morphology and molecular signatures throughout developmental progression. At birth, molecular and cellular transitions occur within the heart, resulting in growth in size and loss of regenerative potential[19-21]. CMs switch from a highly

[1]Department of Developmental Biology, University of Pittsburgh School of Medicine, Pittsburgh, PA 15201, USA. [2]Department of Computational & Systems Biology and Pittsburgh Center for Evolutionary Biology and Medicine, University of Pittsburgh School of Medicine, Pittsburgh, PA 15201, USA. [3]These authors contributed equally: Wei Feng, Abha Bais. ✉e-mail: guangli@pitt.edu

proliferative state to one of hypertrophy, the remodeling of the coronary vascular network results in neonatal-specific population of Vas_ECs, and the regenerative potential in the heart sharply decreases by postnatal day (P) 7[22,23].

Single cell mRNA sequencing (scRNA-seq) is a powerful approach to studying heart development at the single cell level. Using Mesp1-based lineage tracing mice, Lescroart et al.[24] isolated and analyzed the cardiac mesoderm cells with scRNA-seq and identified distinct populations of progenitors committed to different cardiac lineages and regions of the heart. Ivanovitch et al.[25] analyzed the cells in the heart fields using a T-based lineage tracing mouse line and found that cardiac progenitors were spatially prepatterned within the primordial streak. Jia et al.[26] profiled the two heart field progenitors after isolation via *Nkx2-5* and *Isl1* expression identifying novel cell populations. From our previous work profiling early staged murine hearts after microdissection into small zones, we identified zone-specific molecular signatures[27]. Parallel to our investigations, DeLaughter et al.[28] profiled heart cells at five stages (E9.5, E11.5, E14.5, P0, and P21) and found temporal-specific genes. Additionally, several studies have profiled single cells and nuclei at neonatal and adult stages to understand heart maturation and regeneration[29–32]. However, the caveat to many of the listed studies is that the number of profiled cells was small, with most cells being CMs, thus limiting downstream analyses for non-CM lineages. The published datasets are useful for studying early heart development or heart regeneration, but they are missing key developmental timepoints and cell lineages crucial to gaining a better understanding.

Generation of a scRNA-seq dataset that captures every major cardiac cell type is crucial, as non-CMs have been reported to play important roles in heart development. The Endo_ECs in the endocardium are known to secrete growth factors like TGFB1, NOTCH, and Neuregulin-1 to promote myocardium trabeculation[33], while Vas_ECs secrete COL15A1 to promote CM proliferation and inhibit hypertrabeculation[34]. Additionally, the epicardium is known as a hub of growth factors secreting Fibroblast Growth Factor (FGF), Retinoic acid (RA), and Wnt, along with many others, to modulate the myocardium and coronary vasculature development[7,35,36]. Most importantly, scRNA-seq has been demonstrated to have the power to identify the interactions between CMs and non-CMs[37,38].

To understand the function of these signaling interactions, a detailed comparison of their roles under normal and abnormal conditions is important. Wilms' tumor 1 (*Wt1*) and T-Box Transcription Factor 18 (*Tbx18*) are two critical transcription factors in heart development. They are highly expressed in epicardial cells but have been shown to express in other cardiac lineages at early stages[39–42] as well. Homozygous *Wt1* null mice die after E14.5 due to heart deficiency[43], and *Tbx18* mutants die within 24 h of birth as a result of skeletal and respiratory failure[44]. Heart development in *Tbx18* mutants is controversial and has been reported to either result in no defects of any cardiac cell lineages or severe defects in vasculature development[45,46]. A detailed analysis of *Tbx18* mutants at the single-cell level will help elucidate these findings. The single-cell assessment of *Wt1* and *Tbx18* mutants will provide insights into the function of the epicardium throughout heart development.

In this work, we use a multiplexing strategy (MULTI-seq)[47] to profile 72 samples in CD1 mice (18 stages) and 68 samples in C57BL/6 mice (17 stages) with the preservation of stage and chamber identities. Through extensive bioinformatic analysis, we identify stage, chamber, and cell cycle phase-specific gene signatures in each cell lineage. We also identify cell lineage- genes that outperform their current lineage markers on their specificities. Furthermore, we uncover the cell type- and stage-specific enrichment of congenital heart disease (CHD)-associated genes. Lastly, we analyze cardiac cell interactions by assessing ligand and receptor expression and use *Wt1* and *Tbx18* mutants to understand the function of epicardium-derived growth

factors during heart development. Our scRNA-seq dataset and the associated transcriptional features we observe will be invaluable to the understanding of cardiovascular development.

## Results

### ScRNA-seq profiling heart cells at embryonic and neonatal stages

To profile single cell samples at high multiplexity, we used MULTI-seq (Fig. 1A). MULTI-seq is based on cell membrane staining with a lipid modified oligo (LMO) that hybridizes to sample-specific DNA barcodes (Table S1)[47]. Stained samples were pooled and loaded into the 10X Genomics Chromium for single cell isolation and polyA-based reverse transcription. The endogenous mRNA and barcoded oligos were reverse transcribed and amplified together but separated for library generation. After sequencing, the barcode information was used to assign gene reads to each sample (Fig. 1A).

To assess if MULTI-seq can correctly assign gene reads back to each sample, we isolated mouse hearts at E18 and P1 and split them by chambers. We then stained the eight samples with MULTI-seq barcodes and pooled them for scRNA-seq. Computational sample demultiplexing (see "Methods"; Fig. S1) classified ~71% of the total population as "singlets," defined as cells expressing a single MULTI-seq barcode (desired); ~8% as "multiplets," defined as cells expressing more than one MULTI-seq barcode; the rest as "negatives," defined as cells that do not exhibit any MULTI-seq barcode expression (Fig. S2A). After quality control, filtering and normalization (Fig. S2B–F), expression analyses of the atrial CM gene *Sln* and ventricular CM gene *Myl2* showed that *Sln* + CMs were assigned to atrial samples (LA and RA), and *Myl2* + CMs were assigned to ventricular samples (LV and RV) (Fig. S3A, B). These results indicated that MULTI-seq could correctly profile cardiac cells.

Next, we used MULTI-seq to profile heart samples from multiple stages from E9.5 to P9 (Fig. S4). These stages included the major heart developmental points, including four-chamber formation, coronary vascular formation, birth, and loss of cardiomyocyte regeneration (Fig. 1B). To do this, we first analyzed chamber-specific characteristics at ages E9.5 to P3. To assess the impact of loading cell numbers on the scRNA-seq results, we used three cell numbers (5k, 10k, and 25k) in the MULTI-seq analysis. Second, we profiled CD1 hearts from P2 to P9 with the four chambers separated. Third, as described before, we profiled the CD1 hearts from E18.5 and P1. Considering that the CD1 strain has a low penetrance of developing cardiac phenotypes and has been mostly used to study normal heart developmental processes[27,48,49], we also profiled heart cells in C57BL/6—a mouse strain frequently used to model heart defects[50,51]. We analyzed C57BL/6 mouse hearts from E9.5 to P9. In summary, we profiled 72 heart samples with chamber separation at 18 stages in CD1 mice and 68 samples with chamber separation at 17 stages in C57BL/6 mice (Fig. 1B).

After sample demultiplexing based on MULTI-seq barcodes (Fig. S5) and quality control based on sequencing reads, the number of expressed genes, and percentage of mitochondria genes (Fig. S6A, B), we integrated the different batches together and observed no obvious batch differences (Fig. S7A–C). In the CD1 dataset, we captured 65,020 cells consisting of 8987 doublets, 12,313 negatives, and 43,720 singlets. After filtering, 29,001 singlets remained that were distributed throughout the 72 samples (402 cells per sample on average) (Fig. S6C). In the C57BL/6 dataset, we captured 66,171 single cells that included 13,364 doublets, 12,086 negatives, and 40,721 singlets. After filtering, we had 25,605 singlets left across 68 samples (376 cells per sample on average) (Fig. S6C). Through unsupervised clustering analysis of the filtered cells, we found that CD1 and C57BL/6 cells were grouped into 24 and 27 clusters, respectively (Figs. 1C, D, S8). Each cluster has a varied number of cells (Fig. S9, 10).

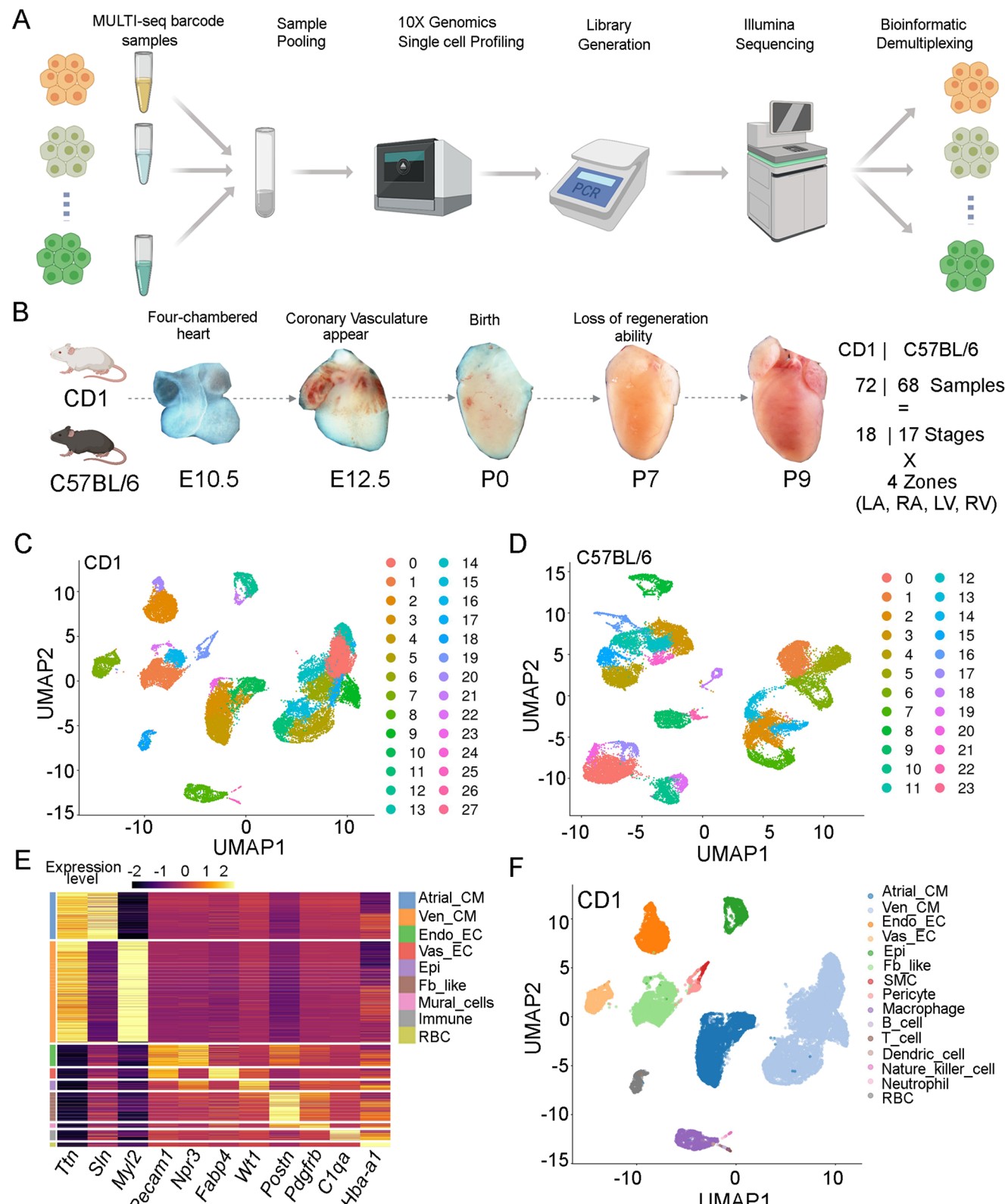

**Fig. 1 | ScRNA-seq analysis of the developing hearts at embryonic and neonatal stages. A** Diagram of the MULTI-seq procedure. Each sample was stained with a unique MULTI-seq barcode before pooling and loading into the 10× Chromium for single cell capturing. The captured single cells were converted into libraries for sequencing and demultiplexing for downstream bioinformatic analyses. **B** Our scRNA-seq datasets include the named mouse strains, developmental stages, and heart zones. Figure 1A and part of Fig. 1B were created with BioRender.com. **C**, **D** Unsupervised clustering of CD1 and C57BL/6 scRNA-seq datasets. **E** The expression pattern of notable CD1 cardiac cell lineage genes. **F** UMAP plot of CD1 cells labeled by cell types.

## Identification of cell types in the scRNA-seq data

Based on our previously published data[27,38], we assigned the cell types in the single cell datasets. We identified nine groups of cells, which could be further separated into 15 sub-groups in the CD1 data (Fig. 1E, F). Based on the expression of pan-CM genes *Ttn*, atrial-specific CM gene *Sln*, and ventricular-specific CM gene *Myl2*, we identified cell clusters representing atrial CMs (Atrial_CMs) and ventricular CMs (Ven_CMs). We also identified clusters corresponding to endocardial ECs (Endo_ECs) and vascular ECs (vas_ECs) based on the expression of pan-EC gene *Pecam1*, endocardial EC gene *Npr3*, and coronary vascular EC gene *Fabp4*. We identified the epicardial cell (Epi) cluster based on the high expression of *Wt1*, *Tbx18*, and *Aldh1a2*. Additionally, we also identified fibroblast-like cells (Fb_like) and mural cells based on the expression of *Postn* and *Pdgfrb*, respectively, and identified immune cells (Immune) and blood cells (RBC) according to the expression of *C1qa* and *Hb1-a1*. Interestingly, the mural cells could be further categorized as smooth muscle cells (SMCs) and pericytes based on the expression of SMC gene *Myh11* and pericyte gene *Pdgfrb*. Immune cells consist of macrophages (*Adgre1*), B-cells (*CD19*), T-cells (*Bcl11b*), dendritic cells (*H2-Oa*), natural killer cells (*Unc13d*), and neutrophils (*S100a9*).

The cell types were not evenly distributed at all stages, with the samples at early stages having more CMs and the samples at neonatal stage having more non-CMs (Fig S11A, C). Consistently, we identified similar cell types and distributions in the C57BL/6 dataset (Figs. S8A, B, C, S11B, D). Integrative analysis of the CD1 and C57BL/6 data revealed high consistency between the two datasets (Fig. S12A, B), which was further supported by the integrative analysis of G1 phased cells in each cell type (Fig. S12C). Furthermore, to identify the subtle molecular differences between strains, we compared the cell type, zone, stage, and cell cycle phase-matched cells and identified a group of genes that expressed differentially between strains (Supplementary Data 1). Interestingly, many genes in this group are pseudo genes and ribosome genes. We have further confirmed the expression of two pseudo genes (*Gm8797*, *Gm10260*) that are differentially expressed between strains using qPCR (Fig. S12D, E). *Gm8797* was predicted to be a ubiquitin B pseudogene, and *Gm10260* was predicted to encode a small ribosomal subunit protein, which, together with the other differentially expressed ribosome genes, suggested that ribosome proteins may play an important role in differentiating the cardiac cells in the two strains. Considering the subtle differences associated with their genetic background, we mainly used the CD1 data for the remaining analyses.

## Identification of stage-specific molecular features

Single cells from each cell type were clustered and colored by stage in Uniform Manifold Approximation and Projection (UMAP) plots. Interestingly, all cell types, except mural and immune cells, were globally organized by developmental stages (Figs. S13Ai, Bi). Specifically, most cells from early embryonic stages clustered together towards one end of the UMAP plots, while those from neonatal stages clustered at the other end while cells from late embryonic stages scattered between the two (Figs. 2A, S14A). Note, in most plots, cells grouped into two main branches attributed to another factor detailed later in the cell cycle section. Next, we colored the cells in the same plots with their pseudotime information and found a correlation with their actual developmental stages (Figs. 2A, S14A).

To identify stage-specific molecular features in each cell type, we first identified the genes that were differentially expressed at the pseudotime stages (Fig. S15). Furthermore, we analyzed gene expression modules consisting of genes with similar expression patterns. Interestingly, we found that most lineage-defining gene modules were enriched at adjacent stages and grouped into two categories: early embryonic or a combination of late embryonic/neonatal stages. Additionally, we found that modules at early stages were enriched with

genes in lineage development and morphogenesis, and the modules at neonatal stages mainly included genes involved in lineage maturation and cellular function. In Atrial_CMs, we identified eight gene modules, with module 8 (M8) being expressed at early embryonic stages with an enrichment of genes in development and morphogenesis pathways like cardiac atrium morphogenesis and muscle cell differentiation (e.g., *Isl1*, *Shox2*, *Bmp2*). Module 2 was expressed at neonatal stages and included genes like *Ttn*, *Pln*, and *Myom2*, which are known to be involved in heart muscle contraction and muscle cell differentiation (Fig. 2Bi and Supplementary Data 2). In Ven_CMs, we found nine gene modules. Module 4 was expressed in early embryonic stages and was enriched with genes in the heart developmental pathways, such as heart morphogenesis and septum morphogenesis (e.g., *Tbx2*, *Tbx3*, *Gata5*, and *Wnt2*). Module 2 in Ven_CMs was expressed during late embryonic and neonatal stages and contained genes enriched for CM maturation-related pathways, such as oxidative phosphorylation and ATP metabolic process electron transport chain (e.g., *Atp5pb*, *Cox7b*, and *Ndufs2*) (Fig. 2Bii). In Epi cells, we identified eight gene modules. Module 8 was expressed at early stages and had genes in the vascular formation and morphogenesis pathways such as kidney vasculature morphogenesis and glomerular capillary formation (e.g., *Tcf21*, *Nrp1*, *Bmp4*, and *Pdgfra*). Module 1 was expressed at late embryonic and neonatal stages with genes from the extracellular matrix-related pathways such as extracellular structure organization and collagen fibril organization (e.g., *Col3a1*, *Cav1*, and *Col1a1*) (Fig. 2Biii). In Vas_ECs, we found seven gene modules. Module 6 was expressed at early stages with genes notably expressed in ribosome biogenesis and rRNA processing (e.g., *Rps2*, *Rps10*, *Rpsa*). Module 3 was expressed at neonatal stages with genes associated with vascular development pathways such as the regulation of angiogenesis and vasculature development pathways (e.g., *Aplnr*, *Cldn5*, *Klf2*, *Klf4*) (Fig. 2Biv). In Fb_like, Endo_ECs, and mural cells, we found the same patterns as described above. However, neonatal immune cells have gene modules for each specific day (Fig. S14B and Supplementary Data 2), which can be attributed to the highly diverse types of immune cells, each known to have their own specific transcriptional profiles.

Furthermore, we analyzed the expression of stage-specific transcription factors (TFs) in each cell type. Like gene modules, we found that the TFs were largely separated into two groups: one highly expressed at early stages from E9.5 to E12.5 and the other at neonatal stages from P0 to P9, which is consistent with the function of TFs in specifying cardiac lineages at early embryonic stage and adapting new environment at neonatal stage. In contrast, few stage-specific TFs were enriched from E13.5 to E18.5, implicating that this period of heart development may be continuous with the previous stages and less changeable. We did not observe a similar TF expression pattern in mural cells, immune cells, and blood cells (Fig. S16 and Supplementary Data 3). Considering that the gene modules and TFs have similar expression patterns (early embryonic or a combination of late embryonic/neonatal stages), we have analyzed if the TFs can regulate the expression of genes in the modules. Through a prediction analysis, we identified a group of regulators that potentially regulate the genes in each module, including several TFs being identified in this study (Supplementary Data 4). For example, we found that *Mef2a*, which was highly expressed at neonatal stage in Atrial_CM, was also predicted to regulate the genes in neonatal-specific gene module (module 2) (Supplementary Data 4). Consistently, *Mef2a* is known as an evolutionarily conserved cardiac core transcription factor, and its mutant mice mostly died in the perinatal stage with cardiac deficiency[52].

## Identification of chamber-specific molecular features

The UMAP plots labeled by zone revealed chamber-specific molecular features (Fig. S13Aii, Bii). Specifically, we observed that atrial CMs from the left and right chambers at late developmental stages were grouped into two distinct populations (LA1, RA1) on UMAP plots (Fig. 3A, S8C)

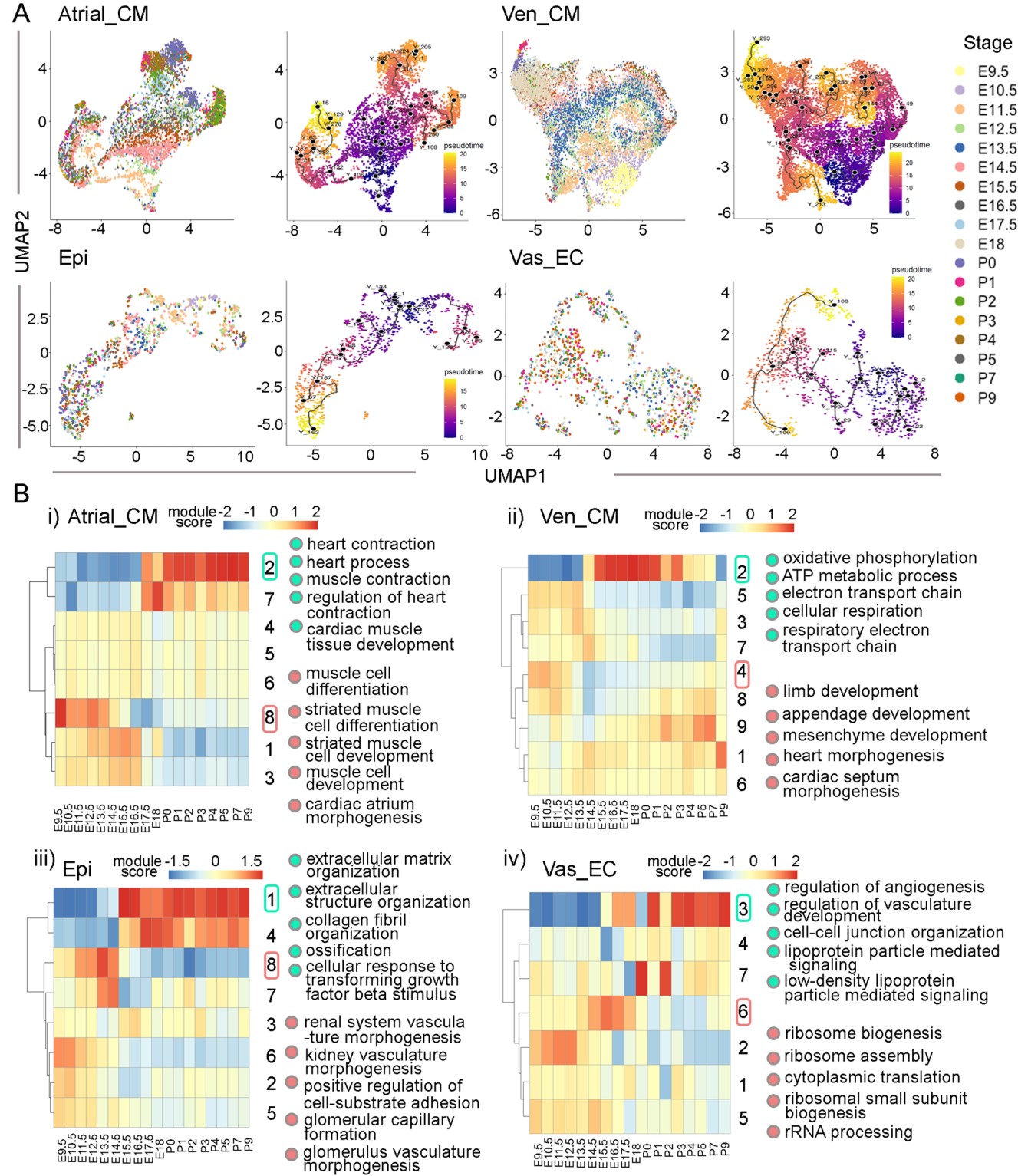

**Fig. 2 | Staged pattern of gene expression modules in each cell type. A** UMAP of single cell lineages labeled with stages and pseudo-time information. **B** Staged gene expression modules and their enriched gene pathways. The color bars represent module scores.

and arose predominantly from stages E16.5 to P7 (Fig. 3B). Differential gene expression analyses identified many genes, including already known chamber-specific gene *Bmp10* and less well-known genes *Ddit4l*, *Adm*, and *Adamts8* (Fig. 3C), to be differentially expressed between the two populations. To confirm the expression pattern of these genes, we performed in situ hybridization and found *Ddit4l* and *Adamts8* to be highly and specifically expressed in the left atrium and

*Bmp10* and *Adm* to be highly expressed in the right atrium (Figs. 3D, S17A, B and Table S2). In Fb_like cells, we found two groups amongst the atrial and ventricular cells (A1, V1) (Fig. 3E). Stage analysis of these cells revealed that they were mostly from E17.5 to P9 (Fig. 3F). Differential expression analysis of these cells identified atrial Fb-highly expressed gene *Sfrp2* and ventricular Fb-highly expressed gene *Mest* (Fig. 3G). The expression patterns of these two genes were further

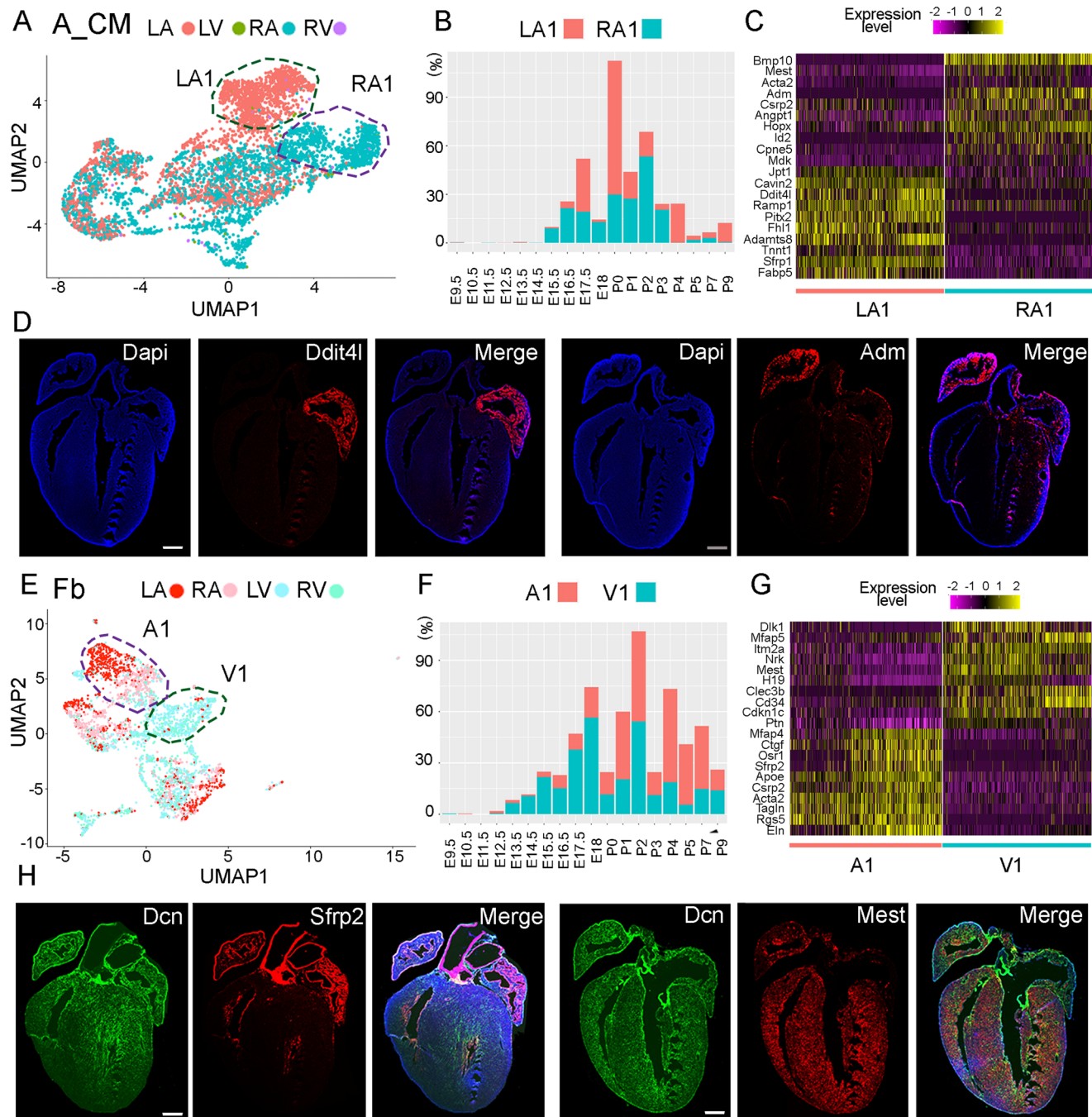

**Fig. 3 | Chamber-specific molecular features of atrial CMs and fibroblasts.**
**A** UMAP of A_CMs labeling the LA and RA-specific populations (LA1, RA1). **B** Stage analyses of LA1 and RA1 cells. **C** Expression heatmap of the top 20 genes differentially expressed in LA1 and RA1 cells. **D** mRNA staining confirmed the LA-specific expression of *Ddit4l* and RA-specific expression of *Adm*. **E** UMAP of fibroblasts revealed atrial and ventricular-specific cell populations (A1, V1). **F** The stage distribution of A1 and V1 cells. **G** Expression of the top 20 differentially expressed genes in A1 and V1 cells. **H** mRNA staining confirmed that *Sfrp2* is specifically expressed in the atrium and *Mest* is specifically expressed in the ventricle. The staining experiments were repeated twice with similar results. Scale bar = 500 µm.

confirmed with in situ hybridization co-stained with the Fb-highly expressed gene *Dcn* (Fig. 3H and Table S2). Additionally, we found that the LA and RA cells in A1 have distinct transcriptional profiles (Fig. 3E). In Endo_ECs, we found an LA- and RA-specific cluster (LA2, RA2) with cells mainly from P0 to P9 (Fig. S18A, B). We also found an LA- and RA-specific cell cluster (LA3, RA3) in epicardial cells, consisting of cells mainly from stages E16.5 to P7 (Fig. S18D, E). Differential gene expression analyses identified genes specifically expressed in LA- or RA-derived cells (Fig. S18C, F). For Ven_CMs, although the LV marker gene *Hand1* and RV marker gene *Pcsk6* were preferentially expressed in LV and RV CMs, respectively, we did not identify zone-specific cell

populations based on the genome-wide gene expression analysis (Fig. S19A, C). This was further supported by the differential gene expression analysis of LV and RV CMs. We identified 19 genes that are highly expressed in LV CMs and 29 genes that are highly expressed in RV CMs (Supplementary Data 5). We also did not observe an enrichment of the septum genes *Irx1* and *Irx2* in specific cell clusters (Fig. S19B). However, we were able to identify an atrioventricular canal (AVC) cluster (cluster 9) that highly expressed the AVC marker genes *Bmp2*, *Rspo3*, *Tbx2*, and *Tbx3*. Note that cluster 9 also contains atrial_CM at E9.5 and E10.5 (Fig. S19D). We can also distinguish between compact and trabecular myocardium cells. The group of trabecular

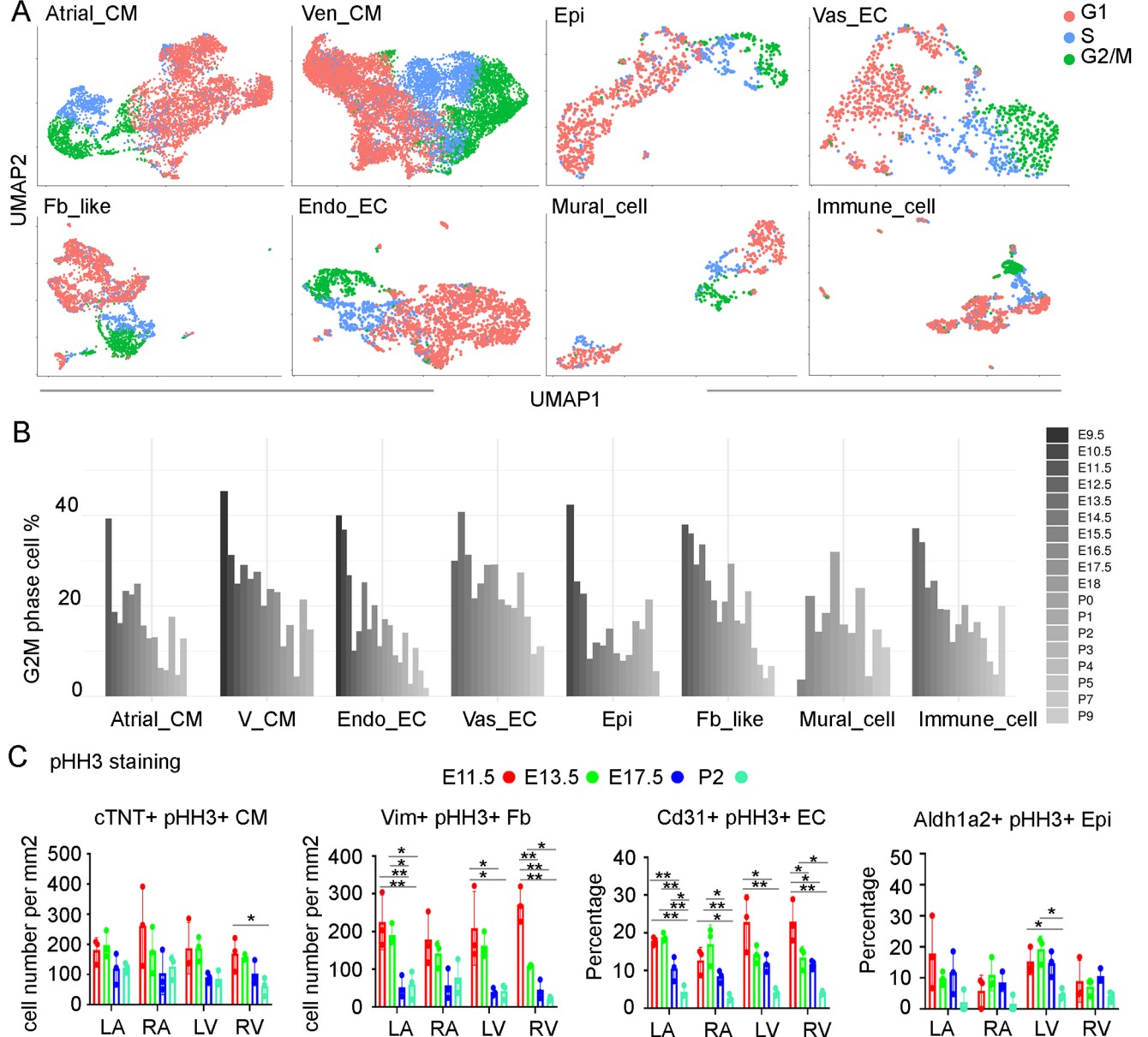

**Fig. 4 | Analysis of cell cycle phases in each cell type. A** UMAP of single cells from each cell type labeled by cell cycle phases. **B** Percentage of G2M phased cells in each cell type declined along the developmental progression. **C** The proportion of pHH3 positive cells in each cell type declined along the developmental progression. $N = 3$ tissue sections were used for the quantifications. ANOVA with Tukey's multiple comparisons was used for the statistical analysis. The error bars represent SD. * and ** indicate significance with $p$ value < 0.05 and 0.01, respectively.

CMs (cluster 4) highly expressed *Bmp10* and *Slit2* while the other clusters, except 4 and 9, expressed *Mycn* and *Hey2*, indicating that they were mainly compact CMs (Fig. S19E). Additionally, we did not find zone-specific cell clusters in Vas_ECs, mural, and immune cells (Fig. S18G–I). In summary, we identified multiple cell types with different transcriptional profiles in LA and RA at late embryonic and neonatal stages. We also showed that *Mest* expresses differentially in LA and RA across several cell types, including Atrial CMs, Fbs, and Endo_ECs (Figs. 3C, G, S18C).

**Identifying cell proliferation changes along the stages**
To understand the cellular heterogeneity within each cell lineage, we labeled the data with their associated cell cycle phases using known cell cycle markers[38,53]. We found that the majority of cells across lineages were grouped by the phases, suggesting their transcriptional profiles were heavily driven by their cell cycle status (Fig. 4A

and Figs. S13Aiii, Biii). Interestingly, we found the two branches from the stage labeled UMAP plots were from two different cell cycle phases, G1 or G2/M and S. Additionally, we observed an overall decline in the percentage of G2/M cells in all cell types along stages (Fig. 4B). However, shared characteristics across cell types did not emerge with a similar analysis with zones (Fig. S20A). Furthermore, when we cleared hearts at four stages (E11.5, E13.5, E17.5, and P2) and stained them with pHH3 to identify the G2/M phased cells, we found that the density of pHH3+ cells also declined from E11.5 to E17.5 but the heart at P2 had slightly higher cell density than the E17.5 heart (Fig. S20Bi–ii), which was further supported by the quantification of pHH3+ cells per zone (Fig. S20C). Lastly, we stained for pHH3 together with lineage marker genes, including cTNT for CM, VIM for Fb, CD31 for EC, and ALDH1A2 for Epi at the same four stages. We found a similar overall decline along the stages in the percentage of G2/M cells in all lineages (Figs. 4C, S21).

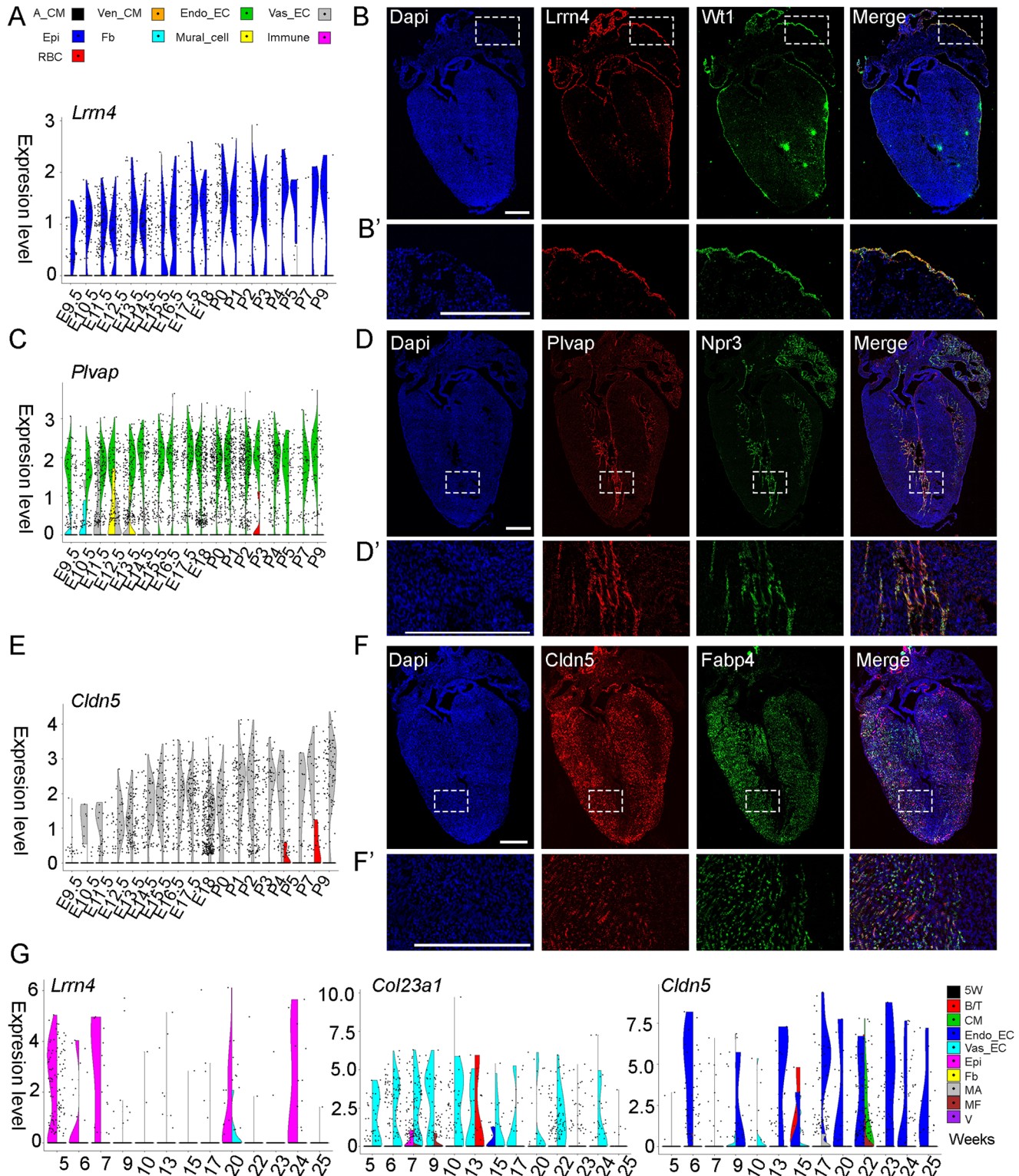

**Fig. 5 | Identification of cell type-specific genes. A** *Lrrn4* is specifically expressed in epicardial cells at all stages. **B**, **B'** In situ RNA staining of *Lrrn4* and known epicardial cell marker *Wt1* confirmed the epicardium-specific expression of *Lrrn4* in P2 hearts. **C** *Plvap* is specifically expressed in Endo_EC at most stages. **D**, **D'** mRNA staining of *Plvap* and known Endo_EC marker *Npr3* showed the Endo_EC specific expression of *Plvap*. **E** *Cldn5* is specifically expressed in Vas_EC at most stages. **F**, **F'** mRNA staining of *Cldn5* and *Fabp4* (a known Vas_EC gene) confirmed that *Cldn5* is specifically expressed in Vas_EC. The staining experiments were repeated in more than three sections with similar results. **G** The expression of *Lrrn4*, *Col23a1*, and *Cldn5* in different human cardiac cell types at fetal stages. Scale bar = 500 μm.

## Discovering cell type-specific marker genes

We reasoned that our datasets could be used to validate the expression pattern of known cell lineage marker genes and identify new ones. While *Wt1* and *Tbx18* are well-known as epicardial lineage markers, our scRNA-seq data revealed them to also be expressed in other cell types at several developmental stages, with *Wt1* being expressed in Vas_ECs and *Tbx18* in Fbs (Fig. S22), maintaining consistency with several previous publications[10,39,42,46]. Other genes,

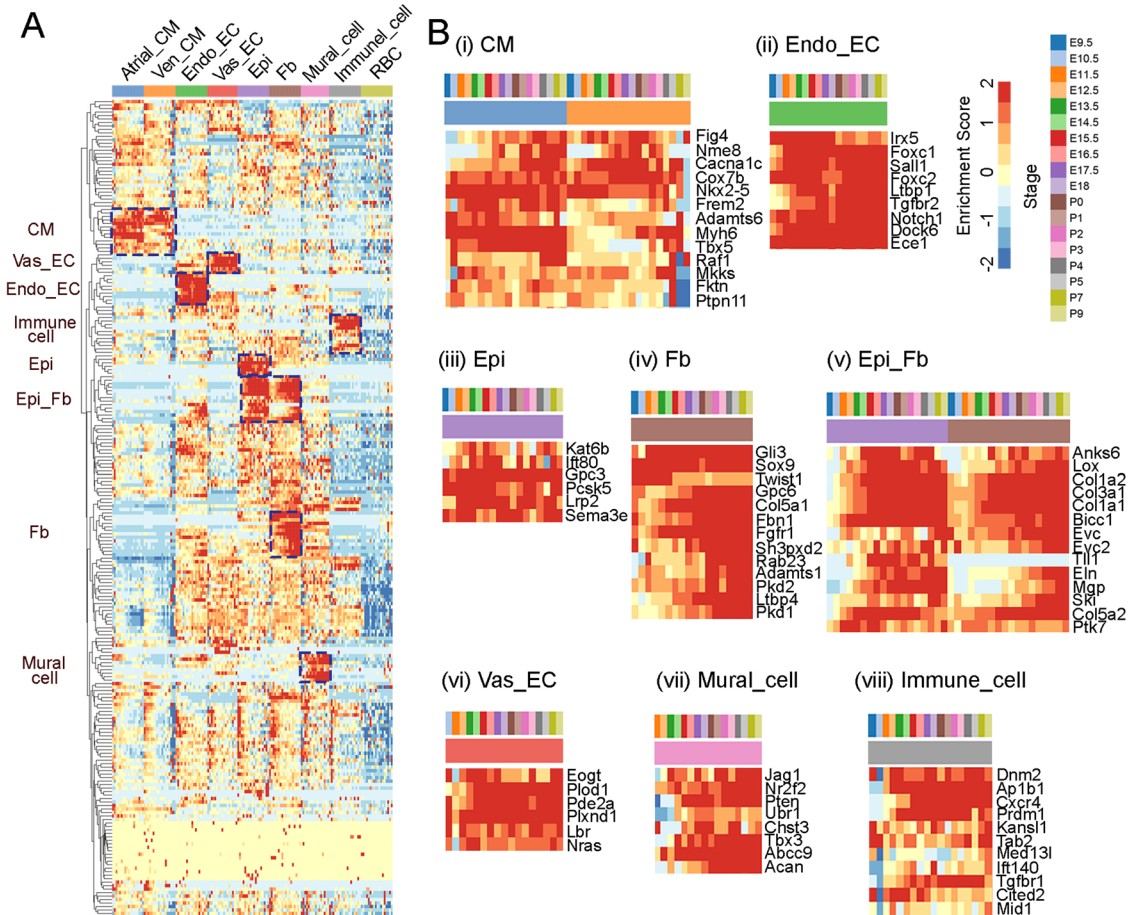

**Fig. 6 | Expression pattern analysis of CHD genes using scRNA-seq data.**
**A** Unsupervised clustering analysis of CHD genes revealed stage and cell type-specific expression patterns. **B** (i–viii) the enlarged expression heatmap of cell type-specific genes. The colors in the color bars represent different stages or cell types. The colors in heatmaps represent gene expression enrichment scores.

including *Npr3* and *Fabp4* were reported to be specifically expressed in Endo_ECs and Vas_ECs, and were frequently used to generate reporter mice to trace their lineages[10,11,23,33]. However, our scRNA-seq data showed that *Npr3* is also expressed in Epi cells at all the analyzed stages, and *Fabp4* is also expressed in Endo_ECs at late embryonic and neonatal stages (Fig. S22). To identify genes that were specifically expressed in one cardiac cell type, we carried out differentially expressed gene analyses between cell types (Fig. S23 and Supplementary Data 6). We found *Lrrn4* and *Apela* to be specifically expressed in Epi cells from E9.5 to P9 (Figs. 5A, S24A); *Col23a1*, *Plvap*, and *Foxc1* to be specifically expressed in Endo_ECs at most stages (Figs. 5C, S24C, E); and *Cldn5* to be specifically expressed in Vas_ECs from E10.5 to P3 (Fig. 5E). To confirm the expression patterns of these less well-known genes, we analyzed their expression at P2 together with the known lineage genes using in situ mRNA hybridization. We found that all genes were expressed at the expected cell types (Figs. 5B, B', D, D', F, F', S24B, B', D, D', F, F' and Table S2). Note that *Cldn5* and *Fabp4* had partially overlapped expression patterns in the staining results, which was probably due to the two genes expressing in partially different Vas_EC populations. We also analyzed the expression patterns of these less well-known genes in human cardiac cells using the published fetal heart scRNA-seq data[54]. We found that *Lrrn4*, *Col23a1*, and *Cldn5* were highly expressed in human epicardial cells, endocardial endothelial cells, and vascular endothelial cells, respectively, across most stages (Fig. 5G). The expression plots also showed that some genes lacked expression at certain stages, likely due to missing cells within the scRNA-seq dataset.

## Expression pattern analysis of CHD-associated genes

Our scRNA-seq dataset includes cardiac lineages from every major heart developmental stage, making it invaluable in assessing expression patterns of genes associated with CHDs. A group of curated known CHD genes, consisting of genes implicated in human CHDs in previous publications and genes shown to cause CHDs in mice, was used to assess their expression patterns with the scRNA-seq data (Supplementary Data 7)[55]. Interestingly, we were able to identify gene clusters that displayed cell type-preferential expression patterns (Fig. 6A). Specifically, we found a cluster of genes to be highly expressed in atrial and ventricular CMs. The scRNA-seq data at different stages further revealed their temporal expression patterns (Fig. 6Bi). The genes in this cluster included *Nkx2-5*, *Myh6*, *Tbx5*, and ten others, which were preferentially related to the CHDs like atrial septal defect (ASD) and ventricular septal defect (VSD). The Endo_EC gene cluster had nine genes, including *Notch1* and *Foxc2* (Fig. 6Bii), which expressed at all stages and were preferentially related to Hypoplastic Left Heart Syndrome (HLHS), Bicuspid Aortic Aalve (BAV), and Tetralogy of Fallot (TOF). Interestingly, the Epi and Fb shared a group of genes besides their own unique gene clusters (Fig. 6Biii–v). The Fb-specific genes included *Sox9*, *Twist1*, and *Fbn1*, which were preferentially related to CHDs with valve defects, such as BAV. However, the Fb and Epi shared genes were related to broad types of CHDs. We also identified gene clusters for Vas_ECs, mural cells, and immune cells (Fig. 6Bvi–viii). The genes expressed in these cell types were also related to broad CHD types. This knowledge will be valuable in understanding the function of these genes in causing the related CHDs in the future.

## Study of cardiac cell communications

We studied the communications between different cardiac cell lineages by analyzing ligand and receptor expression. Through quantification analysis of the ligand-receptor pairs at each stage, we identified a similar number of interactions across the stages with an average of about 250 interactions per stage (Fig. 7Ai). Considering that some interactions were expressed at multiple stages, we also quantified the stages corresponding to each interaction. We found that varied numbers of interactions were expressed at different numbers of stages, with more than 100 interactions being expressed at all 18 stages and about 50 interactions being uniquely expressed in a single stage (Fig. 7Aii). Looking at the zones, we found about 300 interactions in each zone and more than 200 interactions expressed in all four zones (Fig. 7Aiii, iv and Supplementary Data 8). When we made a detailed analysis of the interactions expressed at a single stage, we found more interactions at the postnatal stages than the embryonic stages, and most interactions were between epicardial cell-derived ligands and receptors (Fig. 7B). Our analyses suggest that neonatal non-CMs, like epicardial cells, actively secrete growth factors like BMP2 and BMP4 to regulate postnatal heart growth. Additionally, we successfully retrieved some known interactions in heart development. Specifically, we found that *Nrg1* was expressed in Endo_EC and interacted with *Erbb2* and *Erbb4* in atrial and ventricular CMs (Fig. S25A), which was reported to be critical in the development of myocardium trabeculation[56]. We also found that *Igf1* and *Igf2* were expressed in Endo_ECs and interacted with *Igf1r* and *Igf2r* expressed in a broad number of cell types, including CMs (Fig. S25B), which was known to be important for cell proliferation[57]. However, the interaction analysis did not identify Epi as sending cells in IGF pathways, although both *Igf1* and *Igf2* were found to be expressed in Epi (Fig. S25Biii). Lastly, we found that the Notch ligands *Dlk1*, *Dll4*, and *Jag2*, mainly expressed in Endo_EC and Vas_EC, interacted with *Notch1* from the same cell types (Fig. S25C). Notch signaling in ECs has been known to regulate CM proliferation and differentiation by modulating other signaling pathways, such as *Bmp10* in CMs[58].

We were interested in how non-CMs contribute to ventricular CMs hyperplasic to hypertrophic growth during the fetal to neonatal transitional stages. To do so, we used Nichenet[59] whereby we considered genes that are differentially expressed between E17.5 and P0 Ven_CMs as target genes, receptors expressed in Ven_CMs, and ligands expressed in non-CMs. Interestingly, we found multiple ligands that are expressed in different cell types to have the potential to regulate the same set of genes (Fig. 7Ci, ii and Fig. S26). For example, we found that Fb-expressed ligand MANF, Epi-expressed ligand EFNA5, and EndoEC-derived ligand WNT11 can regulate the same group of genes, including *Ler3*, *Actb*, *Atf3*, *Cited2*, *Per1*, *Pfn1*, *Ctgf*, and *Fhl2*. Besides the genes targeted by all three ligands, MANF and EFNA5 have the potential to regulate another set of genes, including *Eno3*, *Klf9*, *Ranbp1*, and *Rps2*. These genes are from broad pathways associated with the fetal to neonatal transitions and include cell maturation (*Klf9*)[60], circadian rhythm (*Per1*)[61], and metabolic switching (*Eno3*, *Atf3*)[62,63].

To validate the prediction results, we isolated ventricular CMs from E17.5 and newborn (P0–P1) mice and treated them with three growth factors (MANF, EFNA5, and WNT11) to analyze their target genes' expression (Table S3). In the analysis, we selected three genes that were predicted to be regulated by all three ligands (*Atf3*, *Per1*, and *Fhl2*), and four genes that were potentially regulated by MANF and EFNA5 (*Eno3*, *Klf9*, *Ranbp1*, and *Rps2*). The results showed that *Ranbp1* downregulated its expression at E17.5 after MANF treatment and at neonatal stage after MANF and WNT11 treatments. We also found that *Per1* reduced its expression in the EFNA5, MANF, and WNT11-treated samples at neonatal stage (Fig. 7D). These results indicated that the expression of predicted target genes could respond to growth factor treatments and the responses vary between E17.5 and P1 CMs. Lastly, we analyzed if the group of target genes converged to common upstream transcriptional regulators. The prediction analysis identified a group of transcription factors, such as *Gtf2f1*, *Srf*, and *Tbp*, that can potentially co-regulate the genes' expression (Supplementary Data 9).

## Study of the epicardium function with Wt1 and Tbx18 mutants

Epicardial cells are a hub of growth factors that regulate heart development. To understand the communication between epicardial cells and other cell types, we analyzed the interactions between epicardial-derived ligands and receptors from other cell types at each stage. We found that epicardial cells have active communications with other cell types across stages (Supplementary Data 10) and identified stage-specific interactions with a high enrichment at early embryonic and postnatal stages (Fig. 8A).

To better understand the function of the epicardium and its derived growth factors, we used two mouse strains carrying null mutations in the epicardium-expressed transcription factor *Wt1* and *Tbx18* for scRNA-seq. We analyzed both mutants at multiple stages using MULTI-seq to gain a systematic view of their defects. As *Wt1* mutant embryos were reported to die after E14.5, we profiled the hearts at four stages prior: E10.5, E11.5, E13.5, and E14.5 (Fig. S27Ai–iv). Consistent with the previous report[43], we found that the Wt1 mutant embryos at E13.5 and E14.5 had obvious body wall edema, and their hearts had more rounded and bifid apices (Fig. S27Aiii–iv). Histological analysis showed that the *Wt1* mutant had thinner myocardium in the ventricular chambers than the control (Fig. S28A). This was further confirmed by staining analysis of the endothelial cell marker gene CD31 and the myocardium gene cTNT (Fig. S28C). We also profiled *Tbx18* mutants and their littermate control hearts at three stages (E14.5, E15.5, E17.5) (Fig. S27Bi–iii). These stages were relatively later than the *Wt1* mutants as the *Tbx18* mutants were reported to have less severe defects and die after birth (Fig. 8B). We did not identify obvious heart defects in the *Tbx18* mutants based on their morphology (Fig. S27Bi–iii). Additionally, histological analysis of *Tbx18* mutant and control hearts at E15.5 did not identify obvious differences (Fig. S28B). Through further staining analysis of CD31 and cTNT, we did not find significant differences in vessel density and myocardium thickness in the control and the *Tbx18* mutant (Fig. S28D). However, through a whole mount staining analysis of CD31 in *Tbx18* control (*Tbx18*+/−) and mutant (*Tbx18*-/-) hearts at E17.5, we found that the mutant hearts had more ectopic nodules with CD31-positive cells than the control (Fig. S29A). These findings are highly consistent with the observations reported previously[45]. After the standard scRNA-seq processing, we identified the cell types in each sample (Fig. S30A–D). In the *Wt1* mutant and the control samples, we identified a varied number of cells at each stage but the E14.5 samples had the highest cell count (Fig. S31A). Additionally, considering that the Vas_EC starts to develop at E12.5 and becomes one of the main cardiac cell types at E14.5, the remaining analyses were conducted at E14.5. The *Tbx18* mutant and control samples, overall, had a higher number of cells than the *Wt1* samples, and its E17.5 sample had the highest number of cells in each cell type (Fig. S31B).

To understand the defects in the mutants, we analyzed their epicardial cells. Through comparative analyses of the epicardial cells in mutant and control samples at E14.5, we identified a set of genes abnormally expressed in *Wt1* mutant with the downregulation of genes like *Wt1*, *Aldh1a2*, *Rspo1*, and *Tms4f5*, and upregulation of genes like *Tm4sf1* (Fig. 8Ci, Fig. S32A, and Supplementary Data 11). We further confirmed the reduction of ALDH1A2 in Wt1 mutant epicardial cells using immunofluorescence staining. Interestingly, we found that ALDH1A2 expression was mainly reduced in ventricular, but not atrial, epicardium (Fig. S32B). The downregulated genes were enriched in pathways like non-canonical Wnt signaling transduction, and the upregulated genes were from pathways such as monocyte aggregation and lymph vessel development (Fig. S31C). Using the same approach, we also identified abnormally expressed genes in *Tbx18* mutant

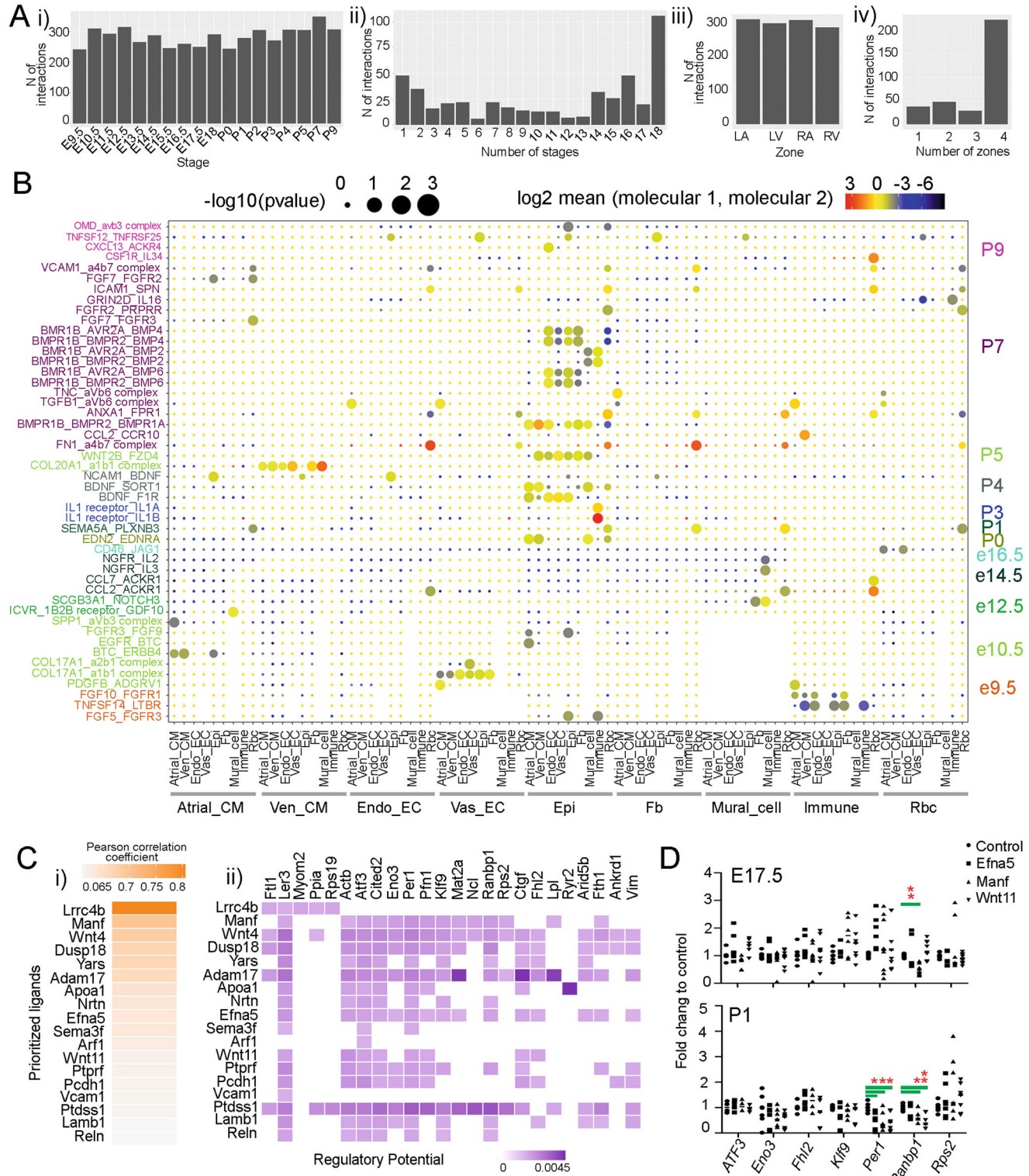

**Fig. 7 | Analyses of ligand-receptor interactions between cardiac cell types. A** (i) The number of interactions at each stage. (ii) Quantification of interactions expressed at different numbers (1–18) of stages. (iii) The number of interactions at each zone. (iv) Quantification of the interactions expressed at different numbers of zones. **B** Expression pattern of ligand-receptor pairs uniquely expressed at one stage. **C** (i) Pearson Correlation Coefficient of prioritized ligands and (ii) The regulatory potential of each ligand on the genes differentially expressed in Ven_CMs at E17.5 and P0. The Pearson correlation coefficient reflects the ability of ligands in

predicting target genes, and the regulatory potential represents the likelihood of a regulation between one ligand and one target gene. **D** Quantification of target genes' expression in E17.5 and newborn mouse CMs after growth factor treatments. The relative expression of each target gene after treatment was normalized to GAPDH and control samples. ANOVA with Dunnett's posthoc test was used for the statistical analysis between each treatment and the control. * and ** indicate significance with $p$ value < 0.05 and 0.01, respectively.

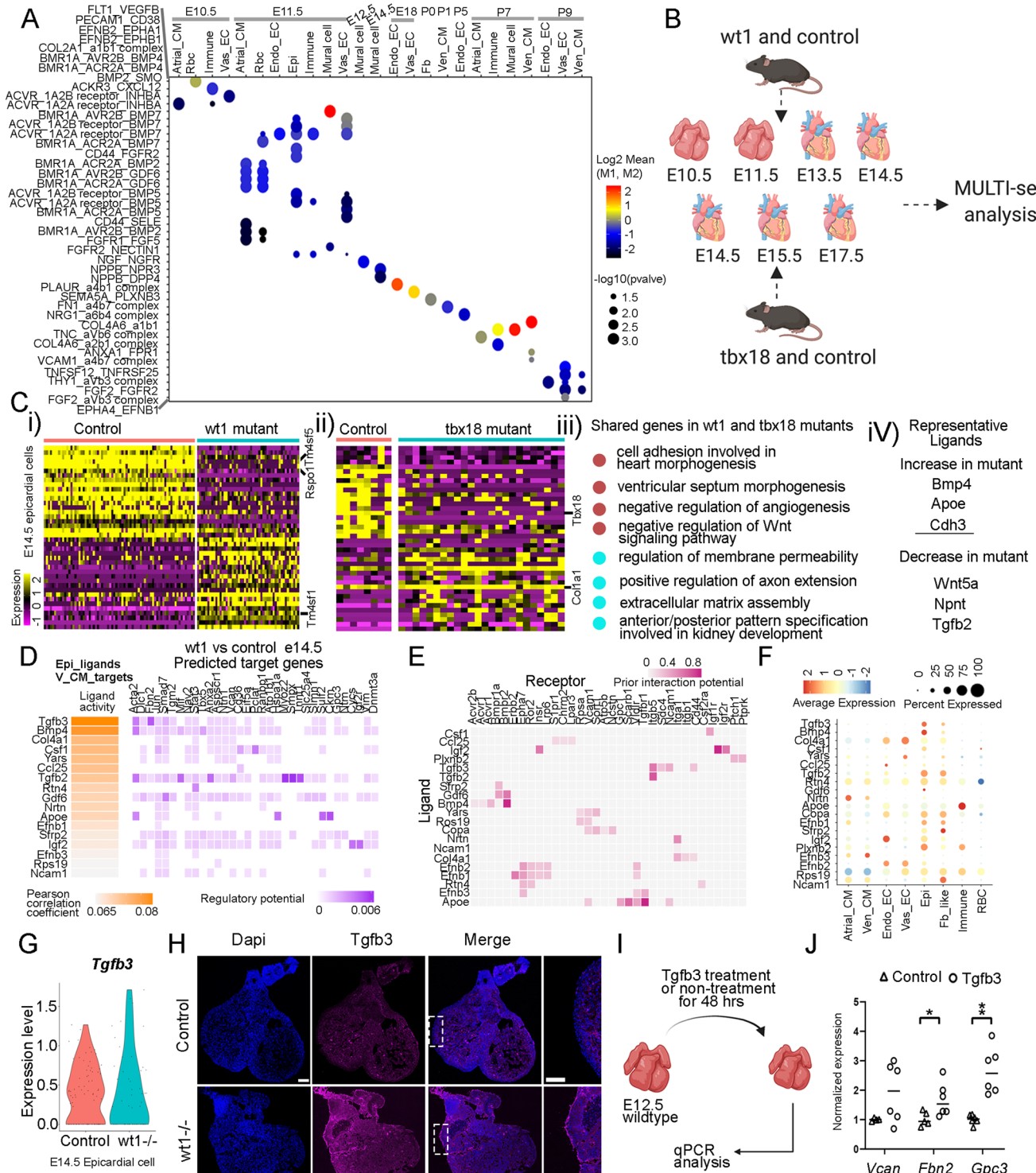

**Fig. 8 | ScRNA-seq analysis of *Wt1* and *Tbx18* mutant hearts at multiple stages.** **A** The stage-unique interactions between ligands from epicardial cells and receptors expressing in other cell types at different stages. **B** Diagram of the profiled samples and their derived developmental stages. This figure was created with BioRender.com. **C** (i, ii) Expression heatmap of the differentially expressed genes in control and *Wt1* or *Tbx18* mutant epicardial cells at E14.5. (iii) Pathway enrichment of abnormally expressed genes shared by *Wt1* and *Tbx18* mutant epicardial cells. (iv) Representative ligands with abnormal expression in *Wt1* and *Tbx18* mutant epicardial cells. **D**–**F** The activity and expression pattern of epicardial cell-derived ligands and their prior interaction potentials with receptors and regulatory

potentials on target genes expression in Ven_CMs at e14.5. **G** ScRNA-seq data revealed the upregulation of *Tgfb3* expression in *Wt1* mutant epicardial cells than controls at E14.5. **H** Immunofluorescence staining confirmed the upregulation of *Tgfb3* in *Wt1* mutant epicardial cells. The staining experiments were repeated twice with similar results. **I, J** Treatment of wildtype hearts with TGFB3 was able to induce the target genes' expression. The diagram was created with BioRender.com. $N = 2$ biologically independent experiments with 3 replicates in each experiment. Student's t-test with two-tailed distribution was used for the statistical analysis. The *p*-value is 0.111 for *Vcan*, 0.042 for *Fbn2*, and 0.0005 for *Gpc3*. Scale bar = 100 μm.

epicardial cells at two stages (E14.5 and E17.5). Downregulated genes include *Tbx18*, and upregulated genes include *Col1a1* (Figs. 8Cii, S29D, and Supplementary Data 12, 14). In general, downregulated genes were enriched for pathways like artery and epithelial tube morphogenesis, and upregulated genes were enriched for pathways such as elastic fiber assembly (Fig. S31E). Next, we compared the abnormally expressed genes in *Wt1* and *Tbx18* mutant epicardial cells and identified a set of genes shared by the two mutants at E14.5 (Supplementary Data 13). Gene ontology analysis of these shared genes revealed an enrichment in multiple pathways, such as the negative regulation of Wnt signaling and extracellular matrix assembly pathways (Fig. 8Ciii). Interestingly, within these shared genes, we found growth factors including *Bmp4*, *Apoe*, and *Cdh3*, which are upregulated in mutants, and *Wnt5a*, *Npnt*, and *Tgfb2*, which are downregulated in the mutants (Fig. 8Civ). Besides epicardial cells, we also analyzed the other cell types and identified differentially expressed genes between the controls and mutants. Overall, we observed more differentially expressed genes in broader cell types in the *Wt1* mutant than the *Tbx18* mutant (Supplementary Data 11, 12), which is consistent with the observation of more severe developmental defects in the *Wt1* mutant than the *Tbx18* mutant. Additionally, we found that Epi, Atrial_CM, and Vas_EC had the most abundant numbers of abnormally expressed genes in the *Wt1* mutant. However, in the *Tbx18* mutant, the cell types are Epi, Endo_EC, and Vas_EC, suggesting that *Wt1* and *Tbx18* have different functions in regulating heart development (Supplementary Data 11, 12).

Furthermore, we compared the ligand-receptor interactions between the controls and mutants to understand how the epicardial cells secrete growth factors to regulate the development of other cell lineages. We were particularly interested in the regulation of ventricular CMs and vascular ECs development by epicardial-derived ligands, as developmental defects in these cell types have been reported in *Wt1* mutants[43]. Through a comparative analysis of *Wt1* controls and mutants, we identified a group of ligands, including, *Tgfb3*, *Bmp4*, and *Col4a1*, that are mainly expressed in epicardial cells and bind to Ven_CM-expressed receptors like *Itgb5*, *Bmpr2*, and *Itga1* to regulate a set of target genes such as *Vcan*, *Fbn2*, and *Gpc3* in Ven_CMs (Fig. 8D–F, Fig. S32C). To confirm if *Tgfb3* expression increased in *Wt1* mutants, we analyzed its expression using scRNA-seq and immunofluorescence staining (Fig. 8H). Next, we treated the cultured embryonic hearts (wildtype) with TGFB3 and found that it can induce the expression of target genes *Vcan*, *Fbn2*, and *Gpc3*, confirming the regulatory interactions between the ligand and target genes (Fig. 8I, J). Importantly, these target genes have been reported to either lead to embryonic heart defects or inhibit cell proliferation, potentially contributing to the heart defects in *W1* mutant[64–66]. Additionally, we also found ligands from epicardial cells such as *Adam17*, *Col4a1*, and *Ccl25* that respectively interacted with Ghr, Ncam1, and Aplnr in Vas_ECs to regulate genes like *Ahnak*, *Ccnd1*, *Gata4*, and *Hes1* in Vas_ECs (Fig. S31F–H). In contrast, similar analyses of *Tbx18* mutant data yielded no genes in Ven_CMs and only a few in Vas_EC that were regulated by ligands from Epi (Fig. S31I–K), reinforcing that *Tbx18* mutants have less severe heart defects than *Wt1* mutants.

## Discussion

In this study, we used a highly multiplexed scRNA-seq strategy to profile cardiac cells at multiple embryonic and neonatal stages. This large dataset can be used as an excellent resource for the study of mammalian heart development. Through analysis of the dataset, we found it to be mainly contributed by three layers of heterogeneity: cell type, stage and zone, and cell cycle phases. Specifically, we found that the gene expression modules and stage-specific transcription factors are highly enriched at early embryonic or neonatal stages but not at late embryonic stages. We also identified distinct transcriptional profiles in the cells of the left and right atrium at late embryonic and neonatal stages. Additionally, we found that the proportion of cells at

G2/M phase declined along developmental progression. Next, we used the dataset for different applications. First, we identified cell lineage markers more specific than the established existing ones. Second, we investigated the expression CHD genes and identified their cell type and temporal expression patterns. Third, we identified the ligand-receptor interactions between different cardiac cell types at each stage. Lastly, we studied epicardium function by analyzing the mutants of two epicardium-expressed transcription factors *Wt1* and *Tbx18* with scRNA-seq and identified abnormalities in different cell types. The results from these analyses confirmed the importance of this dataset.

Previous scRNA-seq analyses of the developing heart were generated from a few stages with low cell numbers, limiting their usage for downstream analyses. Additionally, samples from different stages were profiled separately, which can cause confounding by batch effects. Using sample multiplexing[47], we were able to profile 72 samples from CD1 mice and 68 samples from C57BL/6 mice. Most samples were processed simultaneously and loaded into the single cell pipeline together. Sample overlap between experiments enabled evaluation and showed that our multiplexing strategy efficiently guarded against batch effects. Note that MULTI-seq has the advantage of multiplexing samples, but it can also waste many sequencing reads as some sequenced cells need to be discarded for not having unique MULTI-seq barcodes. Considering that the ventricular are larger than atrial and that hearts at later stages are larger than early stages, our datasets have better coverages in early-hearts and atrial than late-staged hearts and ventricular. Additionally, the hypertrophic growth of ventricular CMs at the neonatal stage makes them too big to fit with the 10X chromium, leading to fewer late neonatal stage ventricular CMs being sampled in our datasets. To profile the ventricular CMs at late neonatal and adult stages, single cell nuclei sequencing would be a better option.

Through stage analyses of the scRNA-seq data, we identified stage-unique gene modules, transcription factors, and ligand-receptor pairs correlating with important heart remodeling occurrences, including heart chamber and coronary vascular development, the adaptation to a normoxic environment after birth, and loss of regenerative potential. It will be interesting to investigate the transcriptional changes occurring at late embryonic stages when the heart is actively growing and preparing for birth. According to the pathway analysis of the genes in module six of Vas_ECs, module one in Atrial_CM, and module six in mural cell (Fig. 2Biv, Fig. S14, and Supplementary Data 2), this stage is enriched with genes associated with ribosome biosynthesis and assembly, which is consistent with active heart growth during this period.

The zone differential expression analyses identified distinct profiles in the left and right atrial in the four cell types atrial_CMs, Endo_ECs, Epi, and Fb (Fig. 3C, G, Fig. S18C, F). These differences are attributed to the developmental sources and physiological environments. Atrial_CMs are known to develop from an embryonic domain different from Ven_CMs at early embryonic stages. Regarding the physiological environments, LA and RA are known to function differently. The LA receives blood from lung circulation (high oxygen blood), and the RA receives blood from venous circulation (low oxygen blood). The RA is closer to the main conduction system components, such as the sinoatrial node and atrioventricular node, than LA, and the LA needs to repress these components' development by expressing genes such as *Pitx2*[67–69]. Considering the transcriptional differences identified at embryonic stages and that the zebrafish atrium has also been reported to have transcriptional differences in the left and right walls[70], the LA-RA differences were most likely caused by the developmental sources rather than physiological environments- but this will require further exploration.

As our dataset covered the major cardiac cell types across 18 stages, we have used it to validate existing cell type markers and identify new, more specific markers. Compared to the traditional methods, such as in situ hybridization, which have been used to

analyze gene expression patterns, scRNA-seq has much better resolution and sensitivity and the potential to be used widely to evaluate the specificity of existing marker genes. However, as scRNA-seq does not preserve spatial information, it should be used in conjunction with in situ hybridization approaches for spatial awareness. Recent breakthroughs pertaining to spatial transcriptomic techniques could be used to generate standard transcriptomic profiles at single cell resolution while preserving the spatial context to validate and identify cell type marker genes[71–73].

Expression pattern analysis of CHD-associated genes using our scRNA-seq data showed cell type and stage identities. This information is basic but critical to understanding the mechanisms of these genes in causing the CHDs. Comparative analyses of CHD gene expression patterns in both mice and humans will be required to evaluate potential mouse models, where only the genes with similar expression patterns in mouse and human cardiac cells can be used to establish models for CHDs. Additionally, as CHD mouse models were mostly developed in inbred mouse strains, the expression pattern of CHD genes in both inbred and outbred strains will be a valuable reference when developing CHD models. As our datasets did not cover the early developmental stages like the cardiac mesoderm stage, it has limitations in assessing the expression pattern of certain CHD genes, which are mainly expressed at early staged cells such as cardiac progenitors.

Our scRNA-seq data showed that *Wt1* and *Tbx18* are not only highly expressed in epicardial cells but also expressed in other cell types, including Vas_ECs and Fbs, consistent with published observations[39,42]. We used mouse mutants of the two genes to study the epicardium function by focusing on the interactions between epicardial-derived ligands and receptors expressed in other cell types. Consistent with previous reports, we found that the *Wt1* mutants had defects in multiple cell types, including Epi, Ven_CMs, A_CMs, and Vas_ECs. However, the *Tbx18* mutant reports were inconsistent. One study reported that *Tbx18* mutant mice had no defects in any cardiac cell types, while another observed defects in the epicardium and vascular system[45,46]. The differences were thought to be caused by the use of different mouse strains: the first study used *Tbx18* null mice in the NMRI-outbred background and the second study used mice with a 129/C57BL6/J mixed background. Our study used *Tbx18* mutant mice with a Black-Swiss-C57BL6/J mixture background, and our scRNA-seq analysis of the mutants identified defects in Epi and Vas_ECs but not in Ven_CMs, which supports the second described study. Furthermore, our study revealed how the epicardium-derived growth factors influenced the transcriptional differences in other cell types, including Ven_CMs and Vas_ECs in the *Wt1* mutant and Vas_ECs in the *Tbx18* mutant.

## Methods
### Experimental part
**Mouse strains.** All animal experiments in the study were approved by the University of Pittsburgh Institutional Animal Care and Use Committee (IACUC). CD1 and C57BL/6N mice were ordered from Charles River Laboratories. *Wt1* mutant mice were generated by breeding pairs of Wt1-GFPCre mice (Strain No: 010911, The Jackson Laboratory). *Tbx18* mutant mice were generated by breeding pairs of Tbx18-CreERT2 mice (Strain No: 031520, The Jackson Laboratory), followed by outbreeding with C57BL/6 mice for three generations.

**Mouse heart dissection and single cell preparations.** Mouse heart dissection and single cell preparations were carried out as described previously[27,74]. Briefly, mouse embryos were harvested from pregnant dams sacrificed by CO$_2$, and neonatal mice were sacrificed through decapitation. To standardize our scRNA-seq datasets, about 5, 4, and 3 hearts from the stage E9.5 to E14.5, E15.5 to E17.5, and P0 to P9 were collected and anatomically dissected by chamber (left and right atrial and left and right ventricle) in cold PBS. The chambers were separated based on anatomical landmarks, such as the septal groove between the LA and RA and between the LV and RV. The AV canal and ventricular septum were collected as part of the LV samples. *Wt1* mutant and control embryos staged at E10.5, E11.5, E13.5, and E14.5 and *Tbx18* mutant and control embryos staged at E14.5, E15.5, and E17.5 were used. Tissues from the same samples were collected together and dissected into smaller pieces. After two washes with cold PBS (Ca$^{2+}$/Mg$^{2+}$ free), the tissue was digested with 0.25% Trypsin/EDTA (Gibco, 25200056) at 37 °C for 10 min. After, the same volume of 20 mg/mL collagenase A and B (Sigma, 10103578001, 11088807001) was added, and the samples were kept at 37 °C until the tissues were digested completely. Vigorous pipetting was used for the postnatal samples and the digestion time varied between samples. After digestion, postnatal staged LV and RV samples were filtered with a 100 μm cell strainer (Corning, 431752). All samples were then collected by centrifugation at 300 × g for 5 min and suspended with 1 mL HBSS (Ca$^{2+}$/Mg$^{2+}$ free) (Gibco, 14170120). Next, cells were filtered with a 40 μm Flowmi cell strainer (Sigma, BAH136800040) and collected by centrifugation at 300 × g at 4 °C for 5 min. The samples were then counted using Nexcelom Cellometer Auto 2000 after being suspended in 1 mL of HBSS (Ca$^{2+}$/Mg$^{2+}$ free). Less than 5 × 10$^5$ cells per sample were used for MULTI-seq barcoding.

**MULTI-seq barcode staining.** MULTI-seq barcoding was carried out as previously described[47]. Single cell samples were washed twice with 1 mL of PBS (Ca$^{2+}$/Mg$^{2+}$ free) and collected by centrifugation at 300 × g for 5 min at 4 °C. After the second centrifugation step, the cells were resuspended in 180 μl PBS (Ca$^{2+}$/Mg$^{2+}$ free). Each sample was incubated with 20 μL Anchor/Barcode stock solution (2 μM Anchor and 5 μL sample specific MULTI-seq barcode in PBS) on ice for 5 min after gentle pipetting. Another 8 min of incubation on ice was performed after gently pipetting with an additional 20 μl of Co-Anchor stock solution (2 μM Co-Anchor in PBS). After washing with cold 1% BSA in PBS, cells were resuspended in 1 mL cold 1% BSA in PBS, and the cell number was counted for each sample. Cells were collected by centrifugation at 300 × g for 5 min at 4 °C and resuspended in 100 μL cold 1% BSA in PBS. To target the same number of cells across the samples, specific volumes of cells from each sample were combined and resuspended in 50 μL 1% BSA in PBS to determine cell concentration.

### Single cell profiling and libraries generation
Based on cell concentration and the targeted number of cells recovered from the 10× Genomics cell suspension (Chromium Single Cell 3' Reagent Kits v3, CG000183 Rev A), we used specified volumes of sample to prepare the cell suspension mixture and loaded them to the 10× Genomics Chromium. CD1 samples were loaded with different volumes of cells into 5 microfluidic wells to target 5k, 10k, and 25k cells in the E9.5_P3 experiments and targeted for 12.5k and 10k cells in P2_P9 and E18_P1 experiments, respectively. For C57BL/6 samples, we loaded the cells into 4 wells to target 12.5k cells in the E10.5_P4 and P5_P9 experiments. For the mutant samples, we used 25k cells. The targeted cell numbers were selected based on the number of samples to profile in each experiment. To generate the endogenous mRNA libraries, the procedure, including GEM generation, mRNA reverse transcription, endogenous cDNA amplification, and library preparation, was carried out via the 10X Genomics Chromium single cell 3' V3 manual with a few changes in the cDNA amplification step[47,74]. Specifically, a MULTI-seq primer was added to the cDNA amplification reaction mix to amplify both endogenous transcript cDNA and barcode cDNA.

To generate the barcode cDNA libraries, we followed our previously published protocol[74]. Briefly, the supernatant from the cDNA cleanup step with 0.6× SPRIselect (Beckman Coulter, B23318) was further cleaned using 3.2× SPRIselect, 80% ethanol, and diluted in EB buffer. The cDNA concentration was measured and used to generate libraries with the KAPA HiFi HotStart ReadyMix (2X) (Roche, KK2601).

After, we cleaned the libraries with 1.6× SPRIselect and 80% ethanol and eluted them in 25 μl EB buffer.

## Sequencing

All gene expression libraries, unless otherwise noted, were sequenced with Illumina HiSeq X platform. The P2_P9 and E10.5-P4 gene expression libraries were sequenced with Illumina Nova-seq platform. The sequencing platforms were selected based on the total number of libraries to be sequenced in each experiment.

## Single molecular in situ hybridization

Gene expression patterns were analyzed using Proximity Ligation In Situ Hybridization (PLISH) as described previously[38,75]. The hearts at P1 and P3 were briefly fixed with 4% PFA (electron microscopy sciences, 15710S) and embedded in OCT (Sakura, 4583). The embedded tissues were then sectioned and treated with a post-fix medium containing 3.7% formaldehyde and 0.1% DEPC (Sigma-Aldrich, D5758) for 30 min. Afterwards, the sections were hybridized with H probes in Hybridization Buffer (1 M NaTCA, 5 mM EDTA, 50 mM Tris pH 7.4, 0.2 mg/mL Heparin) (Table S1). After circulation ligation and rolling circle amplifications, the sections were hybridized with detection probes conjugated to Cy3 or Cy5 fluorophores. Finally, the stained samples were imaged with confocal microscopy (Leica TSC SP8).

## iDISCO analyses of the developing hearts

iDISCO was performed by following the published protocol[76]. The staged CD1 mouse hearts were fixed in 4% PFA at 4 °C overnight. Then, the hearts were dehydrated using a methanol gradient and incubated with 66% DCM/33% Methanol overnight at room temperature (RT) while shaking. After two washes in 100% Methanol, the samples were treated with 5% $H_2O_2$ in methanol (1 volume 30% $H_2O_2$ to 5 volumes methanol) overnight at 4 °C. After, the hearts were rehydrated with a methanol gradient and washed in PTx.2 (0.2% TritonX-100 in PBS) twice at room temperature. The samples were then incubated with Permeabilization Solution (80% PTx.2, 2.3% of Glycine, 20% DMSO) at 37 °C for $n/2$ days ($n = 1$ for the E11.5 and E13.5 hearts; $n = 2$ for the E17.5 and P2 hearts), followed by incubation with Blocking Solution (84% PTx.2, 6% of Donkey Serum, 10% of DMSO) for $n$ days. Next, the samples were incubated with primary antibody Alex488-pHH3 (Abcam ab197502, 1:100 dilution) at 37 °C in primary antibody solution (PTwH/5%DMSO/3% Donkey Serum) for n days and then incubated with secondary antibody in the antibody solution (PTwH/3% Donkey Serum) for n days. After washing in PTwH overnight, the samples were dehydrated in a methanol gradient and kept in 66% DCM/33% Methanol at RT with shaking for 3 h. Furthermore, the samples were incubated in 100% DCM twice with shaking to wash the Methanol. Finally, the samples were incubated in DiBenzyl Ether and imaged using a 3i Lattice Light sheet microscope. Image analysis was conducted through the imaris software (Oxford Instruments). The apex area in each heart was enlarged to the same magnification, and pHH3+ cells in the area were counted to reflect the cell density in the heart. More than ten section images in each heart were used to quantify the chamber areas and pHH3+ cell percentages.

## Immunofluorescence analysis

The heart sections at different stages were stained with Alex488-pHH3 (Abcam ab197502, 1:500 dilution) together with antibodies against lineage genes: cTNT (Abcam, ab45932, 1:400 dilution), Vimentin (Novus Biologicals, NB300-223SS, 1:500 dilution), CD31 (BD, 550274, 1:500 dilution), and ALDH1A2 (Sigma, HPA010022, 1:500 dilution). The *Wt1* mutant and control hearts at E12.5 were sectioned and stained with antibodies for ALDH1A2 (Sigma, HPA010022, 1:250 dilution), TGFB3 (R&D, MAB243SP, 1:100 dilution), CD31 (BD, 550274, 1:100 dilution), and cTNT (Thermo Fisher, MA512960, 1:100 dilution). The *Tbx18* mutant and control heart sections at E15.5 were stained with CD31

(BD, 550274, 1:100 dilution), and cTNT (Thermo Fisher, MA512960, 1:100 dilution). The *Tbx18* mutant and control hearts at E17.5 were whole mount stained with CD31 (BD, 550274, 1:100 dilution) and nuclei dye TO-PRO3 (Thermo Fisher, T3605, 1:1000 dilution).

## Growth factor treatments of mouse CMs or hearts

Ventricular CMs at E17.5 and P0-P1 were isolated from Myh6-Cre/mTmG mice by FACS and cultured in 24-well plates with pre-coating of 0.1% Gelatin. The cells were cultured in mouse differentiation medium, as reported previously[77], and treated with growth factors WNT11 (R&D, 6179WN010, 200 ng/ml), EFNA5 (R&D, 7396-EA-050, 4 μg/ml), or MANF (R&D, 3748-MN-050, 5 μg/ml) for two additional days. Cells were collected in Trizol (Invitrogen, 15596026) and used for RNA extraction, cDNA reverse transcription, and qPCR. The mRNA was extracted using RNeasy Micro kit (Qiagen, 74104), cDNA was generated with iScript cDNA Synthesis kit (BioRad, 1708891), and qPCR was carried out in 7900HT Fast Real-Time PCR System (ABI) and CFX96 Touch Real-Time PCR Detection System (BioRad). The mouse embryonic hearts at E12.5 were cultured in mouse differentiation medium and treated with TGFB3 (R&D, 243-B3-002/CF) for 48 h before being collected for RNA extraction and qPCR analysis.

## Data analysis part

**Data alignment and cell type analysis**. Alignment and quantification of UMI counts for endogenous genes were performed using the cell-ranger count pipeline of the Cell Ranger software (version 3.1.0). We used the mouse reference genome (GRCm38.p4), transcript annotations from Ensembl (version 84), and arguments --chemistry= SC3Pv3 and --expect-cells as 5000, 10,000, and 25,000 based on the specific library. For sample demultiplexing, we used the R package deMULTI-plex (version 1.0.2)[47], consisting of alignment of the MULTI-seq sample barcode read sequences to the reference MULTI-seq sample barcodes followed by sample classification into doublets and singlets. Multiple quality control (QC) metrics were calculated using the R package scater[78], and cells with total library size ≥ 2000, number of detected genes ≥ 1000 and ≤ 8,000, and ≤ 30% percentage of mitochondrial reads were considered. Within-sample doublets were identified and filtered out using the approach described in Feng et al.[79], yielding a total of 29,001 cells for CD1 samples and 25,605 cells for the B6 samples. Figure S7 shows the final number of cells after each stage of QC and filtering. For each sample, we created Seurat[80] objects and processed them with the standard Seurat v4.0 workflow, involving normalization with "LogNormalize," followed by variable feature selection with "vst," and scaling. Individual sample Seurat objects were merged into a single Seurat object, and the top principal components that cumulatively explain >80% of variance in the data were used for batch correction using Harmony[81].

We performed cell type annotation using a top-down approach based on the expression of a panel of lineage genes published by us and others (Fig. 1E, Fig. S8A)[27,38,82]. We first identified cell clusters that broadly resembled major cell types, including cardiomyocytes (*Ttn*+), endothelial cells (*Pecam1*+), epicardial cells (*Wt1*+), fibroblast-like cells (*Postn*+), immune cells (*C1qa*+) and blood cells (*Hba-a1*+). Additional markers further separated putative atrial (*Sln*+) and ventricular cardiomyocyte cell populations (*Myl2*+) as well as endocardial (*Npr3*+) and vascular endothelial cell populations (*Fabp4*+). To take advantage of the granularity offered by the scRNAseq data and further investigate the heterogeneity of major cell types, we analyzed each of the major cell types separately using the same approach as before, including normalization, identifying variable features, dimension reduction, batch correction, and unsupervised clustering. For each subset analysis, we also used cluster-specific markers identified using the FindAllMarkers function in Seurat and performed GO term enrichment analyses of the top 40 markers using gProfiler2[83]. Together, these

enabled us to investigate small outlying subclusters in each subset analysis, excluding possible contaminated cells and assigning more specific cell types. This led to the exclusion of 3 clusters each in atrial cardiomyocytes (clusters 11, 12, and 14), ventricular cardiomyocytes (11, 13, and 14), endocardial endothelial cells (7, 13, and 14), vascular endothelial cells (4, 5, and 9), epicardial cells (5, 7, and 10), 2 clusters in fibroblast-like cells (14 and 17), 4 clusters in immune cells (7, 10, 11, 13), and 1 cluster in blood (cluster 1). Furthermore, the iterative analyses of fibroblast-like cells enabled us to identify and annotate smooth muscle cells (cluster 9) and pericytes (7 and 13). Subset analyses of immune cells yielded macrophages (clusters 0-6), T cells, B cells, natural killer cells, neutrophils, and dendritic cells (clusters 8, 9, 12, 14, and 15, respectively). All downstream analyses were carried out on the cells after removing the clusters marked for exclusion in the individual cell type subsets. Zone-specific genes were identified using the FindMarkers function in Seurat. Finally, Seurat objects from the two strains CD1 and B6 were integrated using Harmony[81], correcting for both batch and strain-associated differences.

The *Wt1* and *Tbx18* mutant and control samples were analyzed with the same workflow as described above for CD1 and C57BL/6 samples. Differentially expressed genes between the mutant and control were identified and plotted using the FindMarkers and DoHeatmap functions in Seurat v4.0.

**Monocle analyses.** We used Monocle (v3)[84] with Harmony batch-corrected PCs for trajectory analyses. For each cleaned (and re-clustered) cell type-specific dataset, we used the same number of PCs as in the original Seurat analysis. Additionally, we used Monocle3 to study stage-specific sets of coregulated genes. For each cleaned cell type-specific subset, we used Monocle from scratch, whereby we selected the top PCs explaining 80% or more of the variance to preprocess the object, followed by Monocle's approach for aligning cells from different batches ("align_cds"), clustering and learning principal graph from the reduced dimension space using reverse graph embedding. We then performed differential expression testing using the graph-auto-correlation analyses using "graph_test" with the principal graph to look for genes that vary between clusters. Monocle 3 runs UMAP on the genes (as opposed to the cells) and then groups them into modules using Louvain community analysis (details can be found on this website: https://cole-trapnell-lab.github.io/monocle3/docs/differential/?q=gene+module#gene-modules).

**Transcription factor expression analysis.** To identify stage-specific transcription factors, we first identified the stage-specific genes in each cell type using FindAllMarkers in Seurat with the default settings. Next, we found the transcription factors in these genes using a complete list of mouse transcription factor genes, which was downloaded from the Mouse Genome Informatics database (http://www.informatics.jax.org/mgihome/GO/project.shtml) with the Gene Ontology term 'DNA binding transcription factor activity'[85]. Finally, we calculated the enrichment score of each transcription factor in each cell type and plotted the top candidates in R.

**Cell cycle phase analysis.** The cell cycle phase of each single cell was assigned as described previously[38]. Briefly, we used the CellCycleScoring function in Seurat to score the cell cycle phases based on the expression of a list of canonical marker genes[53]. Additionally, the percentage of G2/M phased cells at different stages and zones was calculated and plotted in R.

**Violin plot of genes expression.** To generate the gene expression violin plots in Figs. 5, S15, and S16, we exported the cell type-specific gene expression matrix from CD1 Seurat v4.0 objects and loaded them into Seurat v3.0. We generated those plots using VlnPlot in Seurat v3.0. Additionally, to plot the candidate genes' expression in human cardiac cells, we used the human fetal heart scRNA-seq data published in 2019[54] and plotted them in Seurat v3.0[86].

**Motif enrichment analysis.** RcisTarget version 1.16.0 was used to analyze the motif enrichments and predict binding transcription factors on the gene lists. Specifically, the target genes from Nichenet analysis (Fig. 7Cii) and the genes in modules were loaded as inputs. All the parameters were used as default[87].

**CHD genes expression analysis.** To analyze the stage and cell lineage-specific behavior of CHD-associated genes, we used the supplemental data Table S2 of Jin et al.[55]. Briefly, we converted the human genes in the list of curated known CHD genes into mouse genes (233 in total). We looked at the enrichment of each gene in each cardiac cell type and stage using the R package AUCell[87], with a heatmap drawn with the R package ggplot2[88](Figs. 6B, S18, S19). To analyze the CHDs that each gene was associated with, we used the information from the CHD associated risk factors knowledgebase (http://www.sysbio.org.cn/CHDRFKB/).

**Ligand–receptor interaction analysis.** To identify ligand–receptor interactions across stages and zones for each cell type, we analyzed relevant subsets of the CD1 data with CellPhoneDB v3[89], with a *p*-value threshold of 0.2 and number of threads 10, and default parameters for the rest. The significant mean value of all interactive partners (log2) and enrichment *p*-values (−log10) retrieved from the CellPhoneDB outputs were plotted as dotplots in R. CellChat (Version 1.5.0)[90] with the default settings was used to analyze and plot the interactions of Nrg, Igf, and Notch signaling pathways.

Next, we used the R package NicheNet[59] to analyze potential downstream regulation by ligand-receptor pairs of their target genes. Briefly, to study the regulation of genes that are differentially expressed between E17.5 and P0 in Ven_CM, we defined these as the genes of interest and stages E17.5 and P0 as the conditions. We considered Ven_CM cells as receiver cells and cells from other cell types as sender cells, and followed the Nichenet pipeline with default parameters. Similar analyses were performed to study the regulation defects in *Wt1* and *Tbx18* mutant cells, with the difference that Epi cells were considered as senders and all other cell types were considered as receiver cells.

**Reporting summary**
Further information on research design is available in the Nature Portfolio Reporting Summary linked to this article.

## Data availability
ScRNA-seq data from this study have been deposited into the Gene Expression Omnibus (GEO) database under the accession number GSE193346. The processed data has also been deposited to the UCSC cell browser and can be accessed via this link (https://cells-test.gi.ucsc.edu/?ds=mouse-dev-heart). The list of transcription factors was downloaded from the Mouse Genome Informatics database (http://www.informatics.jax.org/mgihome/GO/project.shtml) with the Gene Ontology term 'DNA binding transcription factor activity'.

## Code availability
The main code used to generate the results in this study was provided in the supplement (Supplementary Software 1) and deposited into Zenodo under https://doi.org/10.5281/zenodo.7411556. Other codes are available from the authors upon request.

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

## Acknowledgements

We thank David M. Patterson and Christopher S. McGinnis from Dr. Zev J. Gartner lab for their kind supply of MULTI-seq reagents and experimental and data analysis suggestions. We also thank Dr. Alan Watson from the Center for Biologic Imaging at the University of Pittsburgh for his assistance with imaging the cleared hearts. We thank Valerie Miller from the Rangos Flow Cytometry Core at UPMC Children's Hospital for her help in isolating the CMs. We thank Dr. Michael Tsang for his input on editing the manuscript. We also thank Shirley Chang for helping with cryosectioning and immunostaining some samples and Varun Nair for

helping to quantify pHH3+ staining in the iDISCO experiment. This work was supported by the CMRF grant from the University of Pittsburgh to G.L. and R00HL133472 and DP2HL163745 from the NIH to G.L.

## Author contributions

W.F. and G.L. designed the experiments; W.F. and G.L. performed the scRNA-seq experiments; A.B., H.H., and G.L. designed and performed scRNA-seq data analysis; S.J. and G.L. performed the single molecular in situ hybridization and heart tissue clearing experiments; C.C. performed the qPCR experiments; J.X. bred the mice; W.F., A.B., H.H., C.R., D.K., and G.L. prepared the manuscript; All the authors edited the manuscript.

## Competing interests

The authors declare no competing interests.
