## [Peer Review File · Nature Communications]

Single-cell transcriptomic analysis identifies murine heart molecular features at embryonic and neonatal stagesEditorial Note: Parts of this Peer Review File have been redacted as indicated to remove third-party material where no permission to publish could be obtained.

REVIEWER COMMENTS

Reviewer #1 (Remarks to the Author):

Feng et al have generated a large single cell transcriptomics dataset to investigate transcriptional principles in the mouse heart from embryonic to neonatal stages. The dataset provides a useful resource for the discovery of stage, cell type and region specific markers. In addition, the authors have studied Tbx18 and Wt1 mutant hearts using the same strategy to obtain new insights into the distinct functions of these regulators. A number of points need to be addressed.

1. The authors used multiplexing strategy in both CD1 and BL/6 mice. This effort seems not to be fully exploited in their manuscript. The authors should extend the comparison between these datasets given the variability in penetrance of cardiac phenotypes in BL/6 versus outbred strains. How different are the two datasets? It seems most differences are in ventricular cardiomyocytes from Fig S9. Are there differences in cell populations or is this driven by differences in transcriptional levels of certain genes? Can the authors experimentally validate any differences between CD1 and BL/6?

2. It seems surprising that the analysis does not pick up regional differences between left and right ventricular cardiomyocytes. These cardiomyocytes have different origins and markers specific to the right or left ventricular chambers have been identified, including by single cell RNA-seq approaches. It is unclear why this does not emerge in the current work and the authors need to explain this. This argument could be extended to septal enriched (or excluded) genes and genes expressed in compact versus trabecular myocardium.

3. Please clarify the results for Mest. This gene appears to be differentially expressed between the right and left atria, most interestingly in multiple cell types, yet earlier in the same paragraph the authors refer to Mest as ventricular-specific.

4. Please clarify the relative proliferative changes in different cell types. Can the authors validate this by combining cell-specific markers with pHH3 labelling?

5. The significance of the CHD-associated gene analysis is unclear. Can the authors draw additional conclusions from this part of the study?

6. The manuscript would be enhanced if the authors were able to go further in their interesting observation of common genes regulated by different ligands. Can they show these are converging on common upstream transcriptional regulators? Are the mechanisms common to different cell types?

7. Concerning the comparison of Tbx18 and Wt1, does the broader Wt1 phenotype reflect a broader expression pattern? Can the authors provide any insights into the common target genes? Do they consider that these are indirect changes?

8. The authors speculate that physiological environmental differences such as oxygen levels, may dominate over developmental source in determining left versus right transcriptional differences. However this should be modified, as they document differences at early stages before environmental differences would be expected to intervene. Moreover the left and right walls of the zebrafish atrium have been shown to have transcriptional differences, a study that should be referred to (PMID: 29762122).

9. The authors discuss the possibility that ribosome biosynthesis associated gene changes may be shared by cell types other than vascular endothelial cells. Can't the authors answer this query using their existing data?

10. To ensure the utility of this resource, the authors should make their data easily accessible in a

browser format.

11. Please avoid using the term "novel genes" to describe newly identified markers.

12. Is the color Key correct in Figure 3A?

Reviewer #2 (Remarks to the Author):

What are the noteworthy results?

In "The transcriptional principles in murine heart development at embryonic and neonatal stages", Feng et al. report scRNAseq of each of four cardiac chambers at 18 developmental stages. From these data they made the following observations and conclusions:

- gene expression modules and stage-specific transcription factors (TFs) are enriched in early or late developmental stages, but not in middle stages.
- they identified new, more faithful markers of several cardiac cell types
- they found that congenital heart disease (CHD)-associated genes were expressed in multiple different cell types and were not enriched in cardiomyocytes.
- they identified putative ligand-receptor interactions at different stages of development
- they investigated changes in gene expression resulting from mutation of epicardial TFs WT1 or Tbx18.

Will the work be of significance to the field and related fields? How does it compare to the established literature? If the work is not original, please provide relevant references.

- the data set contains 54605 post-filtering cells from 140 samples representing most of heart development and postnatal cardiac maturation. These data will be valuable for future studies of heart development and disease.
- most of the manuscript is descriptive. Informatic analyses are performed to generate hypotheses, but few are experimentally validated.
- few biological insights emerge from the analysis of the data.

Does the work support the conclusions and claims, or is additional evidence needed?

- few biological insights emerge. Some predictions are made but few are experimentally validated.

Are there any flaws in the data analysis, interpretation and conclusions? - Do these prohibit publication or require revision?

- the manuscript contains significant flaws that require revision.
- one significant flaw is the low number of cells sequenced per sample.
- another significant flaw is the analysis of the CHD data. The definition of a CHD-associated gene needs to be carefully considered. The meaning and biological significance of each CHD lesion should be better understood by the authors and incorporated into the manuscript.
- another significant flaw is in the evaluation of the epicardial mutants. Some of the findings contradict extensive developmental biology studies of epicardium.

Is the methodology sound?

- the scRNAseq methodology is sound. The MULTI-seq methodology is a strength of the manuscript as it minimizes batch effect.
- an important weakness of the study is that the number of cells sequenced per sample is relatively low (average 390 post-filter cells per sample).

Major Comments:

1. Figure clarity

The use of many colors that only subtly differ makes many of the figures difficult to interpret (e.g. 2A, 5A, S15, S22). Several heat maps lack color scales (e.g. 6A, S1, S5, S10B, S11). Color scales and graph axes should have numbers rather than qualitative terms like "low" and "high". In most cases numbers should have units. These problems with the figures significantly hindered critical evaluation of several figures.

2. Cell sampling

Cell sampling per tissue is limited to an average of only 390 post-filter cells. This is a sparse dataset. The authors should critically evaluate how incomplete sampling might impact their data analysis and interpretation. This should also be more made more transparent to readers by displaying plots that quantify cell composition and total cells examined per tissue per stage. Did the size of cardiomyocytes influence their sampling, especially in postnatal stages?

For example, "We observed that atrial CMs from left and right chambers at late developmental stages grouped into two distinct populations (LA1, RA1) on UMAP plots (Fig 3A, S8C), and arose predominantly from stages E16.5 to P7 (Fig 3B)." Could the differences have arisen earlier but have been missed due to incomplete sampling? For example, in figure 3 B, none of the atrial CMs from E10.5 were found in the LA1 or RA1 cluster. Examining Figure 2A shows that most of the atrial CMs found in the dataset were recovered after E15.5. Does limited sampling of earlier stages preclude identifying cells from these stages in LA1 and RA1? Conversely, ventricular cardiomyocytes could not be linked to zone-specific cell clusters Is this due to no chamber specific differences, or because most of the ventricular CMs were recovered before P1 (Fig. 2A)?

3. Data QC

Please provide the UMI number, the gene number, and the mitochondrial gene percentage of the cells for each sample in CD1 and C57BL/6 after quality control. Please show UMAP plots labeling zone and stage for Fig 1C,D to clearly show removal of batch effect. Add bar plots to show the fraction of cells from different zones, stages, and samples in each cell cluster.

3. Mouse strains

The authors studied both CD1 and C57BL/6J. There is no information provided about why both strains were studied, why these two strains were selected, or what insights emerge from inclusion of both strains.

4. Gene modules.

There is no information provided about how gene modules were defined. What can be inferred about how stage-specific TFs regulate the gene modules?

5. Cell trajectories.

The relationship of cell stage to cell trajectory is not clear. For example, in Figure 2 A and Figure 2 B, the pseudotime analysis identifying the two most separate positions in pseudotime (dark purple to bright yellow) are not composed of cells from the most separated embryonic time points in the atrium. The ventricular plot is more obvious, and yet, there are populations of E11.5 cells that show almost the maximal separation in pseudotime.

6. Cell type specific marker genes.

Please provide overall DEG heat maps with labels. How do the markers in this study compare to prior publications from your group and others? In validation experiments, provide both low mag and high mag images with known markers for each cell type.

7. CHD-associated genes.

- The authors should consult with a pediatric cardiologist to understand the terminology and their significance. For example, "Aortic atresia and mitral atresia" are subtypes of HLHS, not separate diseases. "Etiology of Fallot with Pulmonary Alteration" should be "Tetralogy of Fallot with Pulmonary Atresia". Understanding of disease labels would allow the authors to better organize this section.
- The whole exome or whole genome sequencing studies of the PCGC should not be confused with Genome Wide Association Studies.
- The analysis presented is problematic because it does not consider which of the genes with variants found in CHD patients have causal variants vs. which genes are innocent bystanders. Inclusion of many genes likely not causal for CHD contaminates the analysis. The authors should use statistical burden testing to only include genes that are likely functionally linked to CHD.
- In considering the significance of this portion of the study, the authors should consider the probability that many types of CHD may be caused by mutation effects in developmental stages or tissues not sampled in this study (e.g. cardiac progenitors).

8. Ligand-receptor pairs.

- there are several ligand-receptor interactions that are known to be pivotal for heart development. For example, IGF2-IGFR, NRG1-ERBB2/4, JAG/DLL-NOTCH. The authors should show that their analysis methods recover a subset of these known interactions.
- the focus on "stage-specific" interactions may be eliminating biologically significant signals, which are likely to persist beyond a single stage.
- the authors attempt to validate the effect of Wnt11, Manf, and Efna5 on downstream genes but for the most part these validation studies were unsuccessful.
- what is the Pearson correlation coefficient in 7Ci and the "regulatory potential" of a ligand in 7Cii.

9. Epicardial mutants.

- at E9.5, epicardial cells are in the proepicardium and have not yet migrated onto the heart. How is it that heart scRNAseq at E9.5 recovered epicardium? Notably this is the stage with the most "epicardium" ligand-receptor interactions.
- mutation of WT1 abolishes epicardium transition to mesenchymal cells and leads to a paucity of cardiac interstitial cells, such as fibroblasts and smooth muscle cells. Why is this not observed in the cell types identified by scRNAseq?
- at least a subset of putative novel ligand-receptor interactions identified in the Wt1 mutant should be validated to be functionally significant and relevant to the Wt1 cardiac phenotype.

Minor comments

1. "However, neonatal immune cells have gene modules for each specific days (S10B, Table S2), which can be attributed to the highly diverse types of immune cells each known to have their own specific transcriptional profiles" Does this imply that different types of immune cells were captured on particular days? If so, is this an artifact or could more work be done to concretely demonstrate this phenomenon. For example, immunostaining of cell markers at different stages.
2. pHH3 quantification in cleared hearts: Although clearing and staining hearts for pHH3+ is a visually attractive way to answer this question, the authors' claim that pHH3+ levels decrease over time is not immediately apparent from the figure, and how cell density differences were accounted for is not described. Quantification in histological sections with cell type specific markers would be preferred. Statistical analysis is needed to evaluate significance of findings.
3. In the section of "Identification of cell proliferation changes along the stages", please show the cell

cycle phases (G1, S, G2M) in the global UMAP plot to tell if the cell cycle phases would drive the cell type identification.

4. The color of 3F should be kept consistent with 3E.

5. Fig. S2 – is this plot before or after filtering – some cells have low gene number and high percent mt. S2B what does the color represent. S2C there is a cluster with low gene number. What do those represent?

6. Fig. S3 – what is the cell cluster number? A map associating cluster number to annotated names is lacking. Why not show the marker expression in all clusters?

7. Fig. S5-S6. The dataset names are not well described in the figure legend and main text.

8. Fig. S7. Why are only selected cell types shown? This figure could provide a lot more information by showing the number of cells from each stage and zone.

9. Fig. S20. Label the ligands and receptors, or provide the data as a excel spreadsheet.

Reviewer #3 (Remarks to the Author):

Review of “The transcriptional principles in murine heart development at embryonic and neonatal stages” by Wei Feng et al.

In this study, MULTI-seq was performed on 68 samples from 17 stages of 4 zones each (RA, LA, RV, LV) in wildtype hearts from C57BL/6 embryos at each developmental day from E9.5 to P3. They also did a similar experiment of 72 samples at 18 stages of 4 zones each ending at P9, in CD1 embryos. From this, they described different cell populations in the heart observed at different developmental and neonatal stages. They identified GO pathways that are specific to the 4 zones of the heart. Next, they showed that there are differences in gene expression in the left atria and right atria as well as atria and ventricles, with validation of selected genes by low resolution in situ analysis. They then examined the cell populations with respect to cell cycle steps (G1, S, G2/M). This was followed by chamber and cell type marker identification. The study then turns to investigating genes identified for sporadic CHD from a WES dataset published by the PCGC. They examined expression levels in heart cell types at different developmental stages. Next, they investigated ligand-receptor communications in different cell types. Finally, they inactivate Tbx18 and separately Wt1 and perform MULTI-seq on multiple stages to identify differences in endocardial gene expression.

Summary: A major strength of the study is that this project provides an extensive resource of scRNA-seq data for the mouse heart from embryonic to postnatal stages. This data has significant value for the scientific community of individuals studying cardiac development and/or regeneration. This is because the development time-period includes P0-P9 for which birth to P7 includes the times where regeneration is possible.

Major criticisms: I think the major concern of this manuscript is that it tries to cover too much territory and at the end, we are left with more questions than insights. Specifically, there are so many different, seemingly disconnected components to this manuscript that we cannot get a sense of a biological story. Further, it is very descriptive while casting a wide net and therefore provides a superficial global view of the cellular components of the four chambers and how they change over time. Some major points:

1) The team used MULTI-seq barcoding of many samples at many stages, but there are only about

500 cells per sample of each zone, after quality control analysis (Fig S7). Therefore, the resolution of the analysis is not very deep, because of the low cell numbers per sample.

2) They performed scRNA-seq of C57BL/6 and CD1, but the rationale for doing independent experiments with these 2 strains wasn't provided. Was the main purpose to provide a replicate?

3) They refer to cell lineages based upon integrating data from the same zone (LA, RA, LV, RV) at different stages, but this isn't evaluation of genetic lineages. More effort to focus on maybe one or two different aspects of the study with deeper analyses would have improved the manuscript. If the goal is to generate an atlas of non-cardiomyocyte lineages per chamber, then perhaps it would have made sense to perform FACS purification of the lineages themselves and then scRNA-seq, per chamber, despite the lesser number of cells per embryo. The concern is the low number of cells included for each sample. They did not provide the number of reads per cell.

4) Supplemental tables with detailed marker genes per sample and differentially expressed gene for the comparisons described in the manuscript are needed for the wildtype data. Also, supplemental tables of the marker genes for the wildtype vs mutant embryos (to complement Table S9 is needed.

5) Visium spatial transcriptomics would nicely complement the data shown in this study because the resolution of this analysis seems similar.

6) For the human sporadic CHD analyses, the published studies filtered genes based upon the top 25% of genes expressed in the mouse heart at E14.5. It was not clear whether in this analysis of the data, whether they included only these filtered genes. Many of the genes found by the PCGC (the consortium from where the data was obtained) were found in one subject with a variant. Basically, more details from which genes were chosen, perhaps even indicating the supplemental tables that these genes were taken from would have been helpful. Also, some deep insights based upon their analysis besides describing the results would have made their study more interesting. One of the top gene-sets found was chromatin modifiers, but these are ubiquitously expressed. There was no mention of these. Secondly, the known CHD genes are those implicated in CHD but not necessarily those found with mutations in the PCGC or other studies. The analyses presented in Figure 6 have the possibility to be really insightful if this was the focus on the manuscript and there were more follow up experiments provided.

7) The ligand and receptor cell communication findings shown in Figure 7 also comes across as descriptive and of low resolution for the analysis. They did attempt to treat ventricular CMs with growth factors but besides listing some genes, the conclusions were not particularly insightful.

8) Same for the studies on Tbx18 and Wt1 mutant scRNA-seq studies. What were the phenotypes found in their hands, some images would have added some depth, even if some aspects have been published.

Minor criticisms:

1) How many embryos were used per experiment and more metrics details are needed to better understand why CD1 samples were loaded with different volume of cells target 5k, 10k, and 25k cells in the E9.5 P3 experiments, and targeted for 12.5k and 10k cells in P2_P9 and E18_P1 experiments. For C57BL/6 samples, they targeted 12.5k cells in the E10.5_P4 and P5_P9 experiments. For the Tbx18 and Wt1 mutant samples they used 25k cells. Some more details to know why these were chosen. Same for sequencing aspect in that how many reads were done for each sample because it seems some were done using the HiSeq and others were done using the Nova-seq platform.

2) Figure legends and Supplemental Figure legends need to provide much more details. For example Fig S5, its not clear what we are looking at.

3) There are many figures and supplements showing UMAPs of unsupervised clustering of cell types, but there were v few images of feature plots of individual genes that were described in the manuscript. Some violin plots were shown but few individual gene feature plots.

4) The group could have done more bioinformatics analysis to generate cell fate trajectories for a subset of their data providing more depth on top of the one image of pseudotime vs real time (Figure 2A). How can this be easily interpreted to gain some new biological insights of genes that drive cell

fate decisions?

5) They refer to Sup Fig. SA2, and Sup Fig SA3 in the Methods in the section, "Data alignment and cell type analysis". Which figure is that?

REVIEWER COMMENTS

Reviewer #1 (Remarks to the Author):

Feng et al have generated a large single cell transcriptomics dataset to investigate transcriptional principles in the mouse heart from embryonic to neonatal stages. The dataset provides a useful resource for the discovery of stage, cell type and region specific markers. In addition, the authors have studied Tbx18 and Wt1 mutant hearts using the same strategy to obtain new insights into the distinct functions of these regulators. A number of points need to be addressed.

Response: We thank the reviewer for the overall positive comments on our study. Below, please find our response to each specific question.

1. The authors used multiplexing strategy in both CD1 and BL/6 mice. This effort seems not to be fully exploited in their manuscript. The authors should extend the comparison between these datasets given the variability in penetrance of cardiac phenotypes in BL/6 versus outbred strains. How different are the two datasets? It seems most differences are in ventricular cardiomyocytes from Fig S9. Are there differences in cell populations or is this driven by differences in transcriptional levels of certain genes? Can the authors experimentally validate any differences between CD1 and BL/6?

Response: We appreciate the reviewer for the comments. We agree that a comparison of the CD1 and C57BL/6 datasets is important in the study. To do that, we first made an integrative analysis of the two datasets. The integrative UMAP plot in Fig S12A (original Fig S9) showed no clear genome-wide differences between the two strain-derived cells. Considering that each dataset has multiple layers of heterogeneities (cell type, stage, zone, cell cycle phase), we reanalyzed the cells after separating them by cell type and cell cycle phase. As shown in Figure S12C, no clear genome-wide strain differences were identified in G1 phased cells in each cell type. Next, we tried to identify differentially expressed genes between the strains. To better capture the genes that differentially expressed between strains, we compared the matched samples (same cell type, same developmental stage, same zone, G1 phase, similar cell number) from both strains. We identified a handful of genes in each cell type that was preferentially expressed in one strain than the other (Table S2). Lastly, we validated the expression pattern of two genes using qPCR (Figure S12D, E). We have described the results in the manuscript on page 4.

2. It seems surprising that the analysis does not pick up regional differences between left and right ventricular cardiomyocytes. These cardiomyocytes have different origins and markers specific to the right or left ventricular chambers have been identified, including by single cell RNA-seq approaches. It is unclear why this does not emerge in the current work and the authors need to explain this. This argument could be extended to septal enriched (or excluded) genes and genes expressed in compact versus trabecular myocardium.

Response: We thank the reviewer for the questions. We'd like to clarify that the cells were grouped on UMAP plots based on their genome-wide genes expression. The cardiomyocytes from the left and right ventricular could express a subset of genes differentially, but those differences are insufficient to drive them into distinct clusters on the UMAP plot (Fig S19C). Additionally, the Violin plots showed that LV CMs expressed more Hand1 (a reported LV CM marker) while RV CMs expressed more Pcsk6 (a reported RV CM marker), but the feature plots showed that these gene-positive cells do not enrich in

specific cell clusters (Fig S19C). Through further differential gene expression analysis of LV and RV CMs, we were able to identify a small group of genes including 19 genes highly expressing in LV CMs and 29 genes highly expressing in RV CMs. We have included these genes in a supplemental table (Table S7).

Following the reviewer's suggestions, we also analyzed the expression of marker genes for atrioventricular canal (AVC), compact and trabecular myocardium, and septum. Interestingly, we found one cluster of cells (cluster 9) that highly specifically expressed the AVC markers such as *Bmp2*, *Rspo3*, *Tbx2*, and *Tbx3* (Fig S19D), suggesting this cluster of cells were AVC CMs. We also found a cluster (cluster 4) highly expressing trabecular myocardium genes, such as *Bmp10* and *Slit2*, but not the compact myocardium genes, *Mycn* and *Hey2* (Fig S19E). These results suggested that cluster 4 consisted of trabecular CMs, and the other clusters other than cluster 4 and 9 consisted of compact CMs. We also analyzed the septum genes *Ir1* and *Ir2* but did not find their enrichments in any particular cell clusters (Fig S19B), suggesting that septum CMs did not have genome-wide transcriptional differences from the other compact CMs. We have also described these findings in the manuscript (page 6).

3. Please clarify the results for *Mest*. This gene appears to be differentially expressed between the right and left atria, most interestingly in multiple cell types, yet earlier in the same paragraph the authors refer to *Mest* as ventricular-specific.

Response: We thank the reviewer for the comment and apologize for the confusion. We found *Mest* was preferentially expressed in ventricular when we compared the transcriptional profiles of atrial and ventricular fibroblasts (Fig 3G). However, we also found *Mest* was preferentially expressed in the right atrial when we compared the transcriptional profiles of the left and right atrial fibroblasts. Together, these results indicated that *Mest* has high, medium, and low expression in the ventricular, right atrial, and left atrial fibroblasts, respectively. This expression pattern has also been confirmed by the in-situ hybridization results in Fig 3H. We have updated the manuscript to make the description more accurate (page 5).

4. Please clarify the relative proliferative changes in different cell types. Can the authors validate this by combining cell-specific markers with pHH3 labelling?

Response: We thank the reviewer for the comment. Our scRNA-seq analysis identified an overall decline in the proportion of G2M phased cells along the stages. To confirm that, we have co-stained pHH3 with cell type-specific marker genes, including cTNT for cardiomyocyte (A_CM, V_CM), CD31 for endothelial cell (Endo_EC, Vas_EC), Vimentin for Fibroblast, and *Aldh1a2* for Epicardial cell at four stages (E11.5, E13.5, E17.5, P2) (Fig 4C, S21). The staining results largely confirmed the reduction of proliferative cells in each cell type along the developmental progressions.

5. The significance of the CHD-associated gene analysis is unclear. Can the authors draw additional conclusions from this part of the study?

Response: We thank the reviewer for the comment. After careful consideration of the related comments from this reviewer and the other two reviewers, we decided to focus our analysis on a

group of curated known Human/Mouse CHD genes (Table S9, and this reference: Jin et al., Nature Genetics, 2017). Through analysis of these genes, we identified gene clusters that displayed cell type-specific expression patterns and also identified their temporal stage enrichments. Additionally, we checked the genes' preferentially associated CHDs by using the information from the CHD associated risk factors knowledgebase (<http://www.sysbio.org.cn/CHDRFKB/>). The knowledge of these genes' expression preferences (cell type and stage) will be important to understanding their functions in CHDs. We have also updated the manuscript with the new analysis results (Fig 6, page 7, 16).

6. The manuscript would be enhanced if the authors were able to go further in their interesting observation of common genes regulated by different ligands. Can they show these are converging on common upstream transcriptional regulators? Are the mechanisms common to different cell types?

Response: We thank the reviewer for raising an interesting point. Using a package called RcisTarget, which predicted upstream regulators based on motif analysis, we identified a group of transcription factors, including Gtf2f1, Srf, and Tbp, that potentially regulate the set of common genes in ventricular CMs (Table S12). Further expression analysis of these transcription factors in scRNA-seq data showed that they are all highly expressed in Ven_CM, enforcing their potential regulation on the target genes. We also identified a group of genes in Vas_EC that were predicted to be regulated by ligands from multiple cell types, suggesting this is a common mechanism shared by different cell types (Letter Figure 1).

Letter Figure 1: The expression patterns of ligands (A) and their regulatory potentials on the genes differentially expressed in Vas_EC at E17.5 and P0 (B). For instance, Bmp6 from Epi and Cyr61 from Fb are predicted to regulate the expression of a group of genes including Cdkn1a, Cirbp, and Ler in Vas_EC.

7. Concerning the comparison of Tbx18 and Wt1, does the broader Wt1 phenotype reflect a broader expression pattern? Can the authors provide any insights into the common target genes? Do they consider that these are indirect changes?

Response: We thank the reviewer for the comments. According to our scRNA-seq data, Wt1 was mainly expressed in epicardial cells but had low expression in Vas_EC; Tbx18 was mainly expressed in epicardial cells but was lowly expressed in Fibroblasts and mural cells (Fig S22). So we don't think the more severe defects in Wt1 mutants are caused by a broader expression pattern of Wt1. Instead, we think that the less severe defects in Tbx18 mutant are probably due to compensation from other T-box genes such as Tbx5, as discussed previously (Plageman Jr, Yutzey, 2004).

Regarding the common targets in both mutants, we think they probably represent the genes that are not compensated by the other T-box genes in Tbx18 mutants. These genes can be directly or indirectly regulated by Wt1 or Tbx18. Additionally, we found Tbx18 expression to be reduced in Wt1

mutant epicardial cells at E14.5, but no Wt1 expression change in Tbx18 mutant epicardial cells, suggesting that Tbx18 may work downstream of Wt1 to regulate the common set of genes.

8. The authors speculate that physiological environmental differences such as oxygen levels, may dominate over developmental source in determining left versus right transcriptional differences. However this should be modified, as they document differences at early stages before environmental differences would be expected to intervene. Moreover the left and right walls of the zebrafish atrium have been shown to have transcriptional differences, a study that should be referred to (PMID: 29762122).

Response: Thanks! We agree with the reviewer on this point and have updated the manuscript on page 11, accordingly. We have also cited the reference paper.

9. The authors discuss the possibility that ribosome biosynthesis associated gene changes may be shared by cell types other than vascular endothelial cells. Can't the authors answer this query using their existing data?

Response: We thank the reviewer for the comment. After carefully analyzing the gene modules in each cell type, we found that module 6 in Vas_EC, module 1 in Atrial_CM, and module 6 in mural cells are all enriched with ribosome biosynthesis-related pathways (Table S3). Additionally, all of these modules are highly expressed at late embryonic stages (Fig S14B), indicating that this is a phenomenon shared by multiple cell types. We have revised the sentences in the manuscript on this point (page 10).

10. To ensure the utility of this resource, the authors should make their data easily accessible in a browser format.

Response: We appreciated the reviewer for the comment. We have submitted our data to UCSC cell browser. We have included the link (<https://cells-test.gi.ucsc.edu/?ds=mouse-dev-heart>) in the manuscript as well (page 16).

11. Please avoid using the term "novel genes" to describe newly identified markers.

Response: Thanks! We have updated the manuscript by replacing "novel genes" with "less well-known genes."

12. Is the color Key correct in Figure 3A?

Response: We thank the reviewer for the comment. We have double-checked the color key in Figure 3A and confirmed that it is correct.

Reviewer #2 (Remarks to the Author):

What are the noteworthy results?

In “The transcriptional principles in murine heart development at embryonic and neonatal stages”, Feng et al. report scRNAseq of each of four cardiac chambers at 18 developmental stages. From these data they made the following observations and conclusions:

- gene expression modules and stage-specific transcription factors (TFs) are enriched in early or late developmental stages, but not in middle stages.
- they identified new, more faithful markers of several cardiac cell types
- they found that congenital heart disease (CHD)-associated genes were expressed in multiple different cell types and were not enriched in cardiomyocytes.
- they identified putative ligand-receptor interactions at different stages of development
- they investigated changes in gene expression resulting from mutation of epicardial TFs WT1 or Tbx18.

Will the work be of significance to the field and related fields? How does it compare to the established literature? If the work is not original, please provide relevant references.

- the data set contains 54605 post-filtering cells from 140 samples representing most of heart development and postnatal cardiac maturation. These data will be valuable for future studies of heart development and disease.
- most of the manuscript is descriptive. Informatic analyses are performed to generate hypotheses, but few are experimentally validated.
- few biological insights emerge from the analysis of the data.

Does the work support the conclusions and claims, or is additional evidence needed?

- few biological insights emerge. Some predictions are made but few are experimentally validated.

Are there any flaws in the data analysis, interpretation and conclusions? - Do these prohibit publication or require revision?

- the manuscript contains significant flaws that require revision.
- one significant flaw is the low number of cells sequenced per sample.
- another significant flaw is the analysis of the CHD data. The definition of a CHD-associated gene needs to be carefully considered. The meaning and biological significance of each CHD lesion should be better understood by the authors and incorporated into the manuscript.
- another significant flaw is in the evaluation of the epicardial mutants. Some of the findings contradict extensive developmental biology studies of epicardium.

Is the methodology sound?

- the scRNAseq methodology is sound. The MULTI-seq methodology is a strength of the manuscript as it minimizes batch effect.
- an important weakness of the study is that the number of cells sequenced per sample is relatively low (average 390 post-filter cells per sample).

Response: We thank the reviewer for reviewing our manuscript and providing a summary. Please find our responses to each specific point below.

Major Comments:

1. Figure clarity

The use of many colors that only subtly differ makes many of the figures difficult to interpret (e.g. 2A, 5A, S15, S22). Several heat maps lack color scales (e.g. 6A, S1, S5, S10B, S11). Color scales and graph axes should have numbers rather than qualitative terms like “low” and “high”. In most cases numbers should have units. These problems with the figures significantly hindered critical evaluation of several figures.

Response: We thank the reviewer for the comments and apologize for the issues with the scale bar and graph axes. We have improved all figures by using more distinct colors (Fig 2A, 5A, S14A, S24), adding scales (Fig S1, S5, S14B, S16), and adding numbers to scales and graph axes (Fig S2, 2A, S8A, 3C, 3G, S18C, S18F).

2. Cell sampling

Cell sampling per tissue is limited to an average of only 390 post-filter cells. This is a sparse dataset. The authors should critically evaluate how incomplete sampling might impact their data analysis and interpretation.

Response: We appreciate the reviewer for the comment. After multiple steps of QC (removing doublets and barcode-negative cells, and filtering the low-quality cells), we recovered 402 high-quality cells per sample in the CD1 dataset and 376 cells per sample in the C57BL/6 dataset (Fig S8C). We agree that these cell numbers per sample are relatively low. However, considering that the hearts at each stage were dissected into four chambers, there are about 1608 cells per heart in the CD1 dataset and 1504 cells per heart in the C57BL/6 dataset. Additionally, as the integrative analysis of the CD1 and C57BL/6 datasets did not identify genome-wide differences between the two datasets (Fig S12A), their cell numbers can be added up when considering the cellular heterogeneities, which means that we have about 3112 cells per stage. However, considering that the ventricular are larger than the atrial, and the hearts at later stages are larger than early stages, our datasets have better coverages in the early staged hearts and atrial than late staged hearts and ventricular. We have added sentences in multiple places in the manuscript to emphasize this point (Page 4, 10).

This should also be more made more transparent to readers by displaying plots that quantify cell composition and total cells examined per tissue per stage.

We have quantified the cellular compositions at each sample and stage (Fig S11).

Did the size of cardiomyocytes influence their sampling, especially in postnatal stages?

As others have reported before, the cardiomyocyte size increased significantly at late neonatal stages through hypertrophic growth and has less chance of being captured by the 10X Chromium. Very few ventricular CMs were isolated after P3 in the CD1 dataset and after P1 in the C57BL/6 dataset, but a significant number of atrial CMs and other cell types were successfully captured at all stages, including P9 in both datasets.

For example, “We observed that atrial CMs from left and right chambers at late developmental stages grouped into two distinct populations (LA1, RA1) on UMAP plots (Fig 3A, S8C), and arose predominantly from stages E16.5 to P7 (Fig 3B).” Could the differences have arisen earlier but have been missed due to incomplete sampling? For example, in figure 3 B, none of the atrial CMs from E10.5 were found in the LA1 or RA1 cluster. Examining Figure 2A shows that most of the atrial CMs found in the dataset were recovered after E15.5. Does limited sampling of earlier stages preclude identifying cells from these stages in LA1 and RA1?

According to the newly generated cellular composition plot (Fig S11A), atrial CMs in the CD1 dataset were captured starting at E11.5. We also captured atrial CMs at E9.5 and E10.5, but they were grouped together with ventricular CMs (and labeled as Ven_CMs) (Letter Figure 2, Fig S19C, page 6). This is likely due to the similar transcription profile between the two cell types at early stages. In figure 3A, unsupervised clustering analysis identified two groups of cells from left and right atrial, respectively. Further analysis showed that the cells in the two groups were from similar stages, suggesting that they were indeed differentiated by zones rather than by stages. We think profiling more atrial CMs at early stages is unlikely to change the results, as currently, we already have profiled a large number of cells at E14.5 (Fig S11A), but they were not enriched in the LA1 and RA1 clusters.

Conversely, ventricular cardiomyocytes could not be linked to zone-specific cell clusters. Is this due to no chamber specific differences, or because most of the ventricular CMs were recovered before P1 (Fig. 2A)?

Regarding ventricular CMs, we have conducted a more systematic analysis and identified an atrioventricular canal (AVC)-specific cell cluster (Fig S19D). We were also able to differentiate the compact and trabecular myocardium cells (Fig S19E). However, we did not identify genome-wide differences between the LV and RV CMs on the UMAP plots (Fig S19C). As our dataset does not have ventricular CMs at late neonatal stages, we are not sure chamber differences exist at those stages. We have described the analysis results in the manuscript (page 5).

3. Data QC

Please provide the UMI number, the gene number, and the mitochondrial gene percentage of the cells for each sample in CD1 and C57BL/6 after quality control. Please show UMAP plots labeling zone and stage for Fig 1C,D to clearly show removal of batch effect. Add bar plots to show the fraction of cells from different zones, stages, and samples in each cell cluster.

Response: Thanks. We have generated plots showing the UMI numbers, gene numbers, and mitochondrial gene percentages (after QC) in each cell from both the CD1 and C57BL/6 datasets (Fig S6A-B). We have also provided plots (Fig S13Ai-ii, S13Bi-ii) labeling zone and stage for Fig 1C and D.

Lastly, we generated bar plots to show the fraction of cells from different zones, stages, and samples in each cell cluster (the sample name has zone and stage information) (Fig S9, S10).

3. Mouse strains

The authors studied both CD1 and C57BL/6J. There is no information provided about why both strains were studied, why these two strains were selected, or what insights emerge from inclusion of both strains.

Response: We thank the reviewer for the question. We have included CD1 and C57BL/6J strains in this study for two main reasons. First, we can increase the number of cells per stage by profiling the cells in two strains. Second, we would like to see if there are any cellular and molecular differences between the strains, considering the two strains have different penetrance of cardiac phenotypes. Although the UMAP plot of the integrated datasets showed no genome-wide transcriptional differences between the strains (Fig S12A), we were able to identify a handful of differentially expressed genes by comparing the cell type, stage, zone, and cell cycle phase-matched cells in the two strains. We further selected two genes to validate with qPCR (Fig S12D, E). We have included the data as a supplemental figure (Fig S12) and described the results in the manuscript (Page 4).

4. Gene modules.

There is no information provided about how gene modules were defined. What can be inferred about how stage-specific TFs regulate the gene modules?

Response: We thank the reviewer for the comment. We have identified the gene modules using Monocle 3, which essentially run UMAP on the genes (as opposed to the cells) and then grouped them into modules using Louvain community analysis (details can be found on this website: <https://cole-trapnell-lab.github.io/monocle3/docs/differential/?q=gene+module#gene-modules>). According to their descriptions, gene modules represent a group of co-regulated genes. We have included these details in the methods section (Page 15).

Regarding the second part of the question, we think that is an interesting point. To test that, we used a package called RcisTarget to predict the common regulators for the genes in each module (Table S5). Through a further comparative analysis, we found that some common regulators are also stage-enriched transcription factors. For example, we found that the transcription factor Mef2a, which was highly expressed at the neonatal stage in Atrial_CM, was predicted to regulate the genes in the neonatal-specific module (module 2). We have included these analysis results in the manuscript (Page 5).

5. Cell trajectories.

The relationship of cell stage to cell trajectory is not clear. For example, in Figure 2 A and Figure 2 B, the pseudotime analysis identifying the two most separate positions in pseudotime (dark purple to bright yellow) are not composed of cells from the most separated embryonic time points in the atrium. The ventricular plot is more obvious, and yet, there are populations of E11.5 cells that show almost the maximal separation in pseudotime.

Response: We apologize for not precisely understanding what the reviewer referred to in Figure 2A and Figure 2B. We think this could be partially because subtly different colors were used to label the cells at adjacent stages in the original plots, as the reviewer had mentioned in the first comment. We have updated the two plots by using more distinct labeling colors and hope the figures are clearer.

We think the pseudotime plots were largely matched with the stage plots except for some outlier cells. Two main factors could cause the outliers. First, the cells at the same stage can be at different maturation statuses and grouped into different pseudotime stages. Second, the same-staged cells at different cell cycle phases can be at different pseudotime stages, as cells remodeled their transcriptional profiles dramatically at different cell cycle phases.

6. Cell type specific marker genes.

Please provide overall DEG heat maps with labels. How do the markers in this study compare to prior publications from your group and others? In validation experiments, provide both low mag and high mag images with known markers for each cell type.

Response: We thank the reviewer for the comment. We have included a heatmap with the overall DEGs among cell types (Fig S23, Table S8). Compared to the previous markers we and others have published, the markers we have identified in this study have better cell type specificity across stages. As our dataset contains cells at 18 stages, we had the opportunity to analyze the expression pattern of lineage genes at all the stages. For example, we found that the previous lineage genes for epicardial cells (*Wt1*, *Tbx18*), endocardial endothelial cells (*Npr3*), and vascular endothelial cells (*Fabp4*) are not very specific (Fig S22). In contrast, the new genes we have identified in this study are lineage-specific at all or most stages (Fig 5A, C, E). These new genes will be great candidates to generate transgenic mice for lineage tracing.

Additionally, we have provided images with low and high magnifications in the validation experiments. We have also included known marker genes in the experiments (Fig 5, Fig S24).

7. CHD-associated genes.

- The authors should consult with a pediatric cardiologist to understand the terminology and their significance. For example, “Aortic atresia and mitral atresia” are subtypes of HLHS, not separate diseases. “Etiology of Fallot with Pulmonary Alteration” should be “Tetralogy of Fallot with Pulmonary Atresia”. Understanding of disease labels would allow the authors to better organize this section.

Response: We thank the reviewer for the suggestion. We have consulted with a pediatric cardiologist at the UPMC children’s hospital on the terminology and significance of CHDs. However, since we have changed this section to focus on the analysis of a group of curated known CHD genes, we will not include these terms in the manuscript anymore.

- The whole exome or whole genome sequencing studies of the PGC should not be confused with Genome Wide Association Studies.

Response: We apologize for the confusion. They are all from a whole exome sequencing study (Jin et al., Nature Genetics, 2017). We have updated the manuscript to keep this consistent.

- The analysis presented is problematic because it does not consider which of the genes with variants found in CHD patients have causal variants vs. which genes are innocent bystanders. Inclusion of many genes likely not causal for CHD contaminates the analysis. The authors should use statistical burden testing to only include genes that are likely functionally linked to CHD.

Response: We appreciate the reviewer for raising this important point and agree on the problem of using a gene list without burden testing. After careful consideration, we decided to focus on the analysis of a group of curated known CHD genes (Table S9; 234 mouse genes), which consists of genes implicated in human CHDs in previous publications and genes shown to cause CHDs in mice (Jin et al., Nature Genetics, 2017).

- In considering the significance of this portion of the study, the authors should consider the probability that many types of CHD may be caused by mutation effects in developmental stages or tissues not sampled in this study (e.g. cardiac progenitors).

Response: We thank the reviewer for the comment. We have discussed the limitations of our dataset in analyzing some CHD genes expressing at early staged cell types, such as cardiac progenitor cells, which are not covered in our dataset (Page 11).

8. Ligand-receptor pairs.

- there are several ligand-receptor interactions that are known to be pivotal for heart development. For example, IGF2-IGFR, NRG1-ERBB2/4, JAG/DLL-NOTCH. The authors should show that their analysis methods recover a subset of these known interactions.

Response: Thanks. We have analyzed the signaling pathway networks, ligand-receptor interactions, and ligand and receptor expression patterns for IGF, NRG, and NOTCH signaling pathways. We found their expression is largely consistent with what has been reported. For example, we identified strong interactions between Nrg1, Erbb2, and Erbb4. Nrg1 was found to be specifically expressed in Endo_EC, and its receptor Erbb2 and Erbb4 to be expressed in atrial and ventricular cardiomyocytes. We have provided the analysis results as a supplemental figure (Fig S25) and described them in the manuscript (page 7) and cited the related references.

- the focus on “stage-specific” interactions may be eliminating biologically significant signals, which are likely to persist beyond a single stage.

Response: We thank the reviewer for the comment. Besides the stage-specific interactions, we have also provided the interactions existing at different numbers of stages as a supplemental table (Table S10).

- the authors attempt to validate the effect of Wnt11, Manf, and Efna5 on downstream genes but for the most part these validation studies were unsuccessful.

Response: We thank the reviewer for the comment. During revision, we repeated the experiments two more times to increase the biological replicates. The combined results revealed significant changes in a few additional samples, but the results are largely consistent with last time. Some genes have different responses (upregulation or downregulation) to the growth factor treatments in the two samples (E17.5, P1); Some genes only respond to the treatment in one sample, and some genes have no responses at all. As these genes are predicted targets and need to be experimentally validated, this analysis indicates that the genes with expression changes in one or two samples are promising targets. However, we should not conclude that the genes with no expression changes in the experiment are not real targets, as the in vitro system has limitations in recapitulating the in vivo signaling cascades and transcriptional regulations.

- what is the Pearson correlation coefficient in 7Ci and the “regulatory potential” of a ligand in 7Cii.

Response: Both terms are from the package NicheNet, which was designed to predict the ligand-receptor pairs that can mediate the gene expression changes in targeting cells (<https://github.com/saeyslab/nichenetr>). The Pearson correlation coefficient reflects the ability of ligands to predict the target genes, and the regulatory potential represents the likelihood of regulation between one ligand and one target gene. We have included the explanation in figure legends to make this clear.

9. Epicardial mutants.

- at E9.5, epicardial cells are in the proepicardium and have not yet migrated onto the heart. How is it that heart scRNAseq at E9.5 recovered epicardium? Notably this is the stage with the most “epicardium” ligand-receptor interactions.

Response: We thank the reviewer for raising this point. We agree that epicardial cells are mostly still in proepicardium at E9.5, but according to the Fig 1G in this literature (Vicente-Steijn et al., 2015. PMID: 26390289; Letter Figure 3), which made a detailed analysis of the epicardial cell marker (Wt1) at early stages, a few epicardial cells were visible in ventricles. As the other cell types, such as fibroblasts, have not yet developed at this early stage, a small number of epicardial cells can also be captured. However, to avoid potential confusions, we have updated the plot of ligand-receptor interactions (Fig 8A) by excluding the epicardial cells at E9.5.

[redacted]

Letter Figure 3: The Wt1 staining image at E9.5 mouse embryos (original Fig 1G in Vicente-Steijn et al., 2015)

- mutation of WT1 abolishes epicardium transition to mesenchymal cells and leads to a paucity of cardiac interstitial cells, such as fibroblasts and smooth muscle cells. Why is this not observed in the cell types identified by scRNAseq?

Response: We thank the reviewer for the comments. We indeed observed a slight reduction of fibroblasts in the Wt1 mutant compared to the control. Specifically, the Fb percentage at E11.5 reduced from 12% in the control sample to 10% in the mutant, and from 29% in the control to 21% in

the mutant at E14.5. We did not calculate the E10.5 and E13.5 samples as the Fb at E10.5 has not yet developed and the cell number is too low at E13.5.

We reason that there are several possibilities why we did not observe a dramatic reduction of Fb in *Wt1* mutant hearts. First, previous observations, such as the ones in this literature (von Gise et al., 2011), were based on immunofluorescence imaging; single cell mRNA-seq may have better sensitivity than imaging-based approaches in detecting cell types, including fibroblasts. Second, fibroblasts are developed from multiple sources, such as epicardial and endocardial endothelial cells. In the *Wt1* mutant, the endocardial endothelial cells can still differentiate into fibroblasts. To confirm that, we have stained E12.5 staged hearts in control and *Wt1* mutant with Vimentin and observed many positive cells in the mutant (Letter Figure 4), suggesting that Fb can still develop in *Wt1* mutant hearts.

Letter Figure 4: Identification of Vimentin-positive cells at E12.5 mouse embryonic hearts in the control and the *Wt1* mutant.

- at least a subset of putative novel ligand-receptor interactions identified in the *Wt1* mutant should be validated to be functionally significant and relevant to the *Wt1* cardiac phenotype.

Response: We thank the reviewer for the comment. By analyzing the scRNA-seq data, we identified the upregulation of *Tgfb3* in the *Wt1* mutant epicardial cell compared to the control epicardial cell. We further validated the expression change using immunofluorescence staining. Next, we treated the cultured hearts with TGF β 3 protein and observed increased expression of target genes, such as *Vcan*, *Fbn2*, and *Gpc3* (Fig 8G-J). These target genes have already been reported to either lead to embryonic heart defects or inhibit cell proliferation, partially contributing to the heart defects in the *Wt1* mutant. For example, *Vcan* was reported to inhibit cell proliferation and to be mainly expressed in trabecular CMs, which relates to the diminished proliferation of compact myocardium in the *Wt1* mutant. We have also described the results in the manuscript (Page 9).

Minor comments

1. “However, neonatal immune cells have gene modules for each specific days (S10B, Table S2), which can be attributed to the highly diverse types of immune cells each known to have their own specific transcriptional profiles” Does this imply that different types of immune cells were captured on particular days? If so, is this an artifact or could more work be done to concretely demonstrate this phenomenon. For example, immunostaining of cell markers at different stages.

Response: We thank the reviewer for the question. We found that more immune cell types were captured at late embryonic and neonatal stages than early embryonic stages, which is consistent with the stage pattern of gene modules. Considering that a small number of cells from most immune cell types except macrophage were profiled, this phenomenon is probably an artifact.

2. pHH3 quantification in cleared hearts: Although clearing and staining hearts for pHH3+ is a visually attractive way to answer this question, the authors’ claim that pHH3+ levels decrease over time is not

immediately apparent from the figure, and how cell density differences were accounted for is not described. Quantification in histological sections with cell type specific markers would be preferred. Statistical analysis is needed to evaluate significance of findings.

Response: We appreciate the reviewer for the question. We have described the method of quantifying pHH3+ cell density in cleared hearts in the manuscript (Page 13). We have also co-stained pHH3 with cell type-specific markers on histological sections. In the results from both experiments we have performed statistical analysis to evaluate the significances (Fig 4C, S20C).

3. In the section of “Identification of cell proliferation changes along the stages”, please show the cell cycle phases (G1, S, G2M) in the global UMAP plot to tell if the cell cycle phases would drive the cell type identification.

Response: Thanks. We have added global UMAP plots labeled with cell cycle phases (Fig S13 A iii, B iii). From these plots, we do not think the cell cycle phases have driven the cell type identifications.

4. The color of 3F should be kept consistent with 3E.

Response: Thanks. As figure 3F has two variants (A1 and V1) and Figure 3E has four variants (LA, RA, LV, and RV), we have selected two similar colors for the two zones in A1 or V1 in Figure 3E to match with the colors in Figure 3F.

5. Fig. S2 – is this plot before or after filtering – some cells have low gene number and high percent mt. S2B what does the color represent. S2C there is a cluster with low gene number. What do those represent?

Response: We thank the reviewer for the comment and apologize for the confusion. The plots are before filtering. We have presented them to show what the data looks like before filtering. We have added the plots after filtering, and from the plots we can see that the clusters with low gene numbers were filtered out. (Fig S2E, F). Additionally, the color in Fig S2B represents the total number of molecules detected in each cell. We have added this information to the figure legend as well.

6. Fig. S3 – what is the cell cluster number? A map associating cluster number to annotated names is lacking. Why not show the marker expression in all clusters?

Response: We apologize for the confusion. We only showed the cardiomyocyte clusters in the original plots. To avoid confusion, we have updated the plots by including all clusters and labeling the CM clusters based on Tnni3 expression (Fig S3).

7. Fig. S5-S6. The dataset names are not well described in the figure legend and main text.

Response: We apologize for the issue. We have added more details to the figure legends and manuscript to improve the descriptions (page 3).

8. Fig. S7. Why are only selected cell types shown? This figure could provide a lot more information by showing the number of cells from each stage and zone.

Response: Thanks. As most plots in the original Fig S7 are redundant with the plots in the new Figure

S9, we only kept the plots with cell numbers at different QC steps, which are now in Fig S8C. In Fig S9 and S10, the number of cells at each stage and zone was provided.

9. Fig. S20. Label the ligands and receptors, or provide the data as a excel spreadsheet.

Response: Thanks! After careful consideration, we decided to remove this supplemental figure, as it represents the relative expression level of ligands and receptors instead of the potential interactions between them.

Reviewer #3 (Remarks to the Author):

Review of “The transcriptional principles in murine heart development at embryonic and neonatal stages” by Wei Feng et al.

In this study, MULTI-seq was performed on 68 samples from 17 stages of 4 zones each (RA, LA, RV, LV) in wildtype hearts from C57BL/6 embryos at each developmental day from E9.5 to P3. They also did a similar experiment of 72 samples at 18 stages of 4 zones each ending at P9, in CD1 embryos. From this, they described different cell populations in the heart observed at different developmental and neonatal stages. They identified GO pathways that are specific to the 4 zones of the heart. Next, they showed that there are differences in gene expression in the left atria and right atria as well as atria and ventricles, with validation of selected genes by low resolution in situ analysis. They then examined the cell populations with respect to cell cycle steps (G1, S, G2/M). This was followed by chamber and cell type marker identification. The study then turns to investigating genes identified for sporadic CHD from a WES dataset published by the PCGC. They examined expression levels in heart cell types at different developmental stages. Next, they investigated ligand-receptor communications in different cell types. Finally, they inactivate Tbx18 and separately Wt1 and perform MULTI-seq on multiple stages to identify differences in endocardial gene expression.

Summary: A major strength of the study is that this project provides an extensive resource of scRNA-seq data for the mouse heart from embryonic to postnatal stages. This data has significant value for the scientific community of individuals studying cardiac development and/or regeneration. This is because the development time-period includes P0-P9 for which birth to P7 includes the times where regeneration is possible.

Major criticisms: I think the major concern of this manuscript is that it tries to cover too much territory and at the end, we are left with more questions than insights. Specifically, there are so many different, seemingly disconnected components to this manuscript that we cannot get a sense of a biological story. Further, it is very descriptive while casting a wide net and therefore provides a superficial global view of the cellular components of the four chambers and how they change over time. Some major points:

Response: We thank the reviewer for reviewing our manuscript and providing the summary. Below, please find our responses to each specific point.

1) The team used MULTI-seq barcoding of many samples at many stages, but there are only about 500 cells per sample of each zone, after quality control analysis (Fig S7). Therefore, the resolution of the analysis is not very deep, because of the low cell numbers per sample.

Response: We thank the reviewer for the comment. After multiple steps of QC (removing doublets and barcode negative cells and filtering the low-quality cells), we recovered 402 high-quality cells per sample in the CD1 dataset and 376 cells per sample in the C57BL/6 dataset (Fig S8C). We agree that these cell numbers per sample are relatively low. However, considering that the hearts at each stage were dissected into four chambers, there are about 1608 cells per heart in the CD1 dataset and 1504 cells per heart in the C57BL/6 dataset. Additionally, as the integrative analysis of the CD1 and C57BL/6 datasets did not identify genome-wide differences between the two datasets (Fig S12A), their cell numbers can be added up when considering the cellular heterogeneities, which means that we have about 3112 cells per stage. However, considering that ventricular are larger than atrial, and the hearts at later stages are larger than early stages, our datasets have better coverages in the early staged hearts and atrial than late staged hearts and ventricular. We have added sentences in multiple places in the manuscript to emphasize this point (Page 4, 10).

2) They performed scRNA-seq of C57BL/6 and CD1, but the rationale for doing independent experiments with these 2 strains wasn't provided. Was the main purpose to provide a replicate?

Response: We appreciate the reviewer for the question. We have included CD1 and C57BL/6J strains in this study for two main reasons. First, we can increase the number of cells per stage by profiling the cells in two strains. Second, we would like to see if there are any cellular and molecular differences between the strains, considering the two strains have different penetrance of cardiac phenotypes. Although the UMAP plot of the integrated datasets showed no genome-wide transcriptional differences between the strains (Fig S12A), we were able to identify a handful of differentially expressed genes by comparing the cell type, stage, zone, and cell cycle phase-matched cells in the two strains. We further selected two genes to validate with qPCR (Fig S12D, E). We have included the data as a supplemental figure (Fig S12) and described the results in the manuscript (Page 4).

3) They refer to cell lineages based upon integrating data from the same zone (LA, RA, LV, RV) at different stages, but this isn't evaluation of genetic lineages. More effort to focus on maybe one or two different aspects of the study with deeper analyses would have improved the manuscript. If the goal is to generate an atlas of non-cardiomyocyte lineages per chamber, then perhaps it would have made sense to perform FACS purification of the lineages themselves and then scRNA-seq, per chamber, despite the lesser number of cells per embryo. The concern is the low number of cells included for each sample. They did not provide the number of reads per cell.

Response: We thank the reviewer for the comments. The goal of this study is to generate an atlas of CMs and non-CMs in developing mouse hearts at embryonic and neonatal stages. We did not use FACS to purify lineage-specific cells for scRNA-seq as that will introduce more biases to the dataset. Because our dataset covered 18 developmental stages with the preservation of chamber identities, it will have great value in understanding the developmental progression of each lineage and the communications between lineages. However, as our dataset has low cell numbers in certain samples, it will have its limitations, as we have specified in response to the reviewer's 1st comment. We have also discussed this limitation in the manuscript (Page 4, 10). Additionally, we have quantified the

cellular compositions at each sample and stage to make it clear to the readers (Fig S11). Finally, we have generated plots showing the UMI numbers, gene numbers, and mitochondrial gene percentages (after QC) in each cell from both CD1 and C57BL/6 datasets (Fig S6A-B).

4) Supplemental tables with detailed marker genes per sample and differentially expressed gene for the comparisons described in the manuscript are needed for the wildtype data. Also, supplemental tables of the marker genes for the wildtype vs mutant embryos (to complement Table S9 is needed).

Response: Thank you. As each sample consists of multiple cell types, the differential expression analysis between samples is difficult to generate meaningful results. Instead, we have identified the marker genes for each cell type (Table S8, Fig S23) and further identified the gene modules and transcription factors along the stages in each cell type (Table S3, S4, Fig 2B, Fig S16). Additionally, we have identified the genes that expressed differentially between controls and mutants (wt1 or tbx18) in each cell type and stage (Table S14 and S15).

5) Visium spatial transcriptomics would nicely complement the data shown in this study because the resolution of this analysis seems similar.

Response: We thank the reviewer for the comment. The Visium spatial transcriptomic data may nicely complement our scRNA-seq data by providing further spatial information, but we do not think the current Visium data can reach single-cell resolution yet. Additionally, generating a new set of spatial transcriptomics data is out of the scope of this study.

6) For the human sporadic CHD analyses, the published studies filtered genes based upon the top 25% of genes expressed in the mouse heart at E14.5. It was not clear whether in this analysis of the data, whether they included only these filtered genes. Many of the genes found by the PCGC (the consortium from where the data was obtained) were found in one subject with a variant. Basically, more details from which genes were chosen, perhaps even indicating the supplemental tables that these genes were taken from would have been helpful. Also, some deep insights based upon their analysis besides describing the results would have made their study more interesting. One of the top gene-sets found was chromatin modifiers, but these are ubiquitously expressed. There was no mention of these. Secondly, the known CHD genes are those implicated in CHD but not necessarily those found with mutations in the PCGC or other studies. The analyses presented in Figure 6 have the possibility to be really insightful if this was the focus on the manuscript and there were more follow up experiments provided.

Response: We appreciate the reviewer for the comments. We agree with the reviewer that the known CHD genes are those implicated in CHDs but not necessarily those found with mutations in the PCGC or other studies. After careful consideration, we decided to focus on the analysis of a group of curated known CHD genes, consisting of genes implicated in human CHDs in previous publications and genes shown to cause CHDs in mice (Jin et al., Nature Genetics, 2017). We have also provided this list of genes as a table in the manuscript (Table S9; 234 mouse genes). Through analysis of these genes, we identified gene clusters that displayed cell type-specific expression patterns and also identified their temporal stage enrichments. Additionally, we checked the genes' preferentially associated CHDs

by using the information from the CHD associated risk factors knowledgebase (<http://www.sysbio.org.cn/CHDRFKB/>) (Fig 6, page 7, 16).

7) The ligand and receptor cell communication findings shown in Figure 7 also comes across as descriptive and of low resolution for the analysis. They did attempt to treat ventricular CMs with growth factors but besides listing some genes, the conclusions were not particularly insightful.

Response: We thank the reviewer for the comment. In the analysis of ligand-receptor communications, we identified the stage and zone enriched interactions, which will serve as a great resource for future experimental validations. Because we are interested in heart remodeling after birth in this study, we selected the ventricular CMs at E17.5 and P1, the two stages right before and after birth. Using an algorithm named NicheNet that can link ligands to target genes, we identified a group of genes that differentially expressed at the two stages and further linked their potential regulatory ligands. Lastly, we experimentally validated the regulatory interactions and identified several promising interactions. Considering these genes are from pathways directly associated with the fetal to neonatal transitions and include cell maturation (Klf9), circadian rhythm (Per1), and metabolic switching (Eno3, Atf3) pathways, we respectively disagree with the reviewer that this part of the study is not insightful.

8) Same for the studies on Tbx18 and Wt1 mutant scRNA-seq studies. What where the phenotypes found in their hands, some images would have added some depth, even if some aspects have been published.

Response: We thank the reviewer for the comment. We have provided images for Tbx18 and Wt1 mutant and control hearts at all analyzed stages (Fig S27). Additionally, through analysis of the scRNA-seq data, we identified a ligand Tgfb3 whose expression increased in Wt1 mutant epicardial cells. We further validated the expression upregulation using immunofluorescence staining. Furthermore, we treated the cultured hearts with TGFB3 protein and observed the induction of target genes such as Vcan, Fbn2, and Gpc3 (Fig 8G-J). The three genes have been reported to lead to congenital heart defects or inhibit cell proliferations, which is tightly related to the diminished CM proliferation in Wt1 hearts. We have added this in the figures and the manuscript (Fig 8G-J, page 9).

Minor criticisms:

1) How many embryos were used per experiment and more metrics details are needed to better understand why CD1 samples were loaded with different volume of cells target 5k, 10k, and 25k cells in the E9.5 P3 experiments, and targeted for 12.5k and 10k cells in P2_P9 and E18_P1 experiments. For C57BL/6 samples, they targeted 12.5k cells in the E10.5_P4 and P5_P9 experiments. For the Tbx18 and Wt1 mutant samples they used 25k cells. Some more details to know why these were chosen. Same for sequencing aspect in that how many reads were done for each sample because it seems some were done using the HiSeq and others were done using the Nova-seq platform.

Response: We thank the reviewer for the comments. Specifically, we collected 5, 4, and 3 hearts from stages E9.5 to E14.5, E15.5 to E17.5, and P0 to P9 and further dissected them by chambers. Regarding

the targeting cell numbers, we first tested three numbers (5k, 10k, and 25k) in the E9.5_P3 experiment. The further integrative analysis revealed no significant differences among the experiments with different targeting cell numbers (Fig S7). In the rest of the experiments, we chose to target cell numbers based on the number of samples to profile in each experiment to reach even numbers per sample. Regarding the sequencing reads, we targeted more than 20K reads per cell based on the recommendations from 10X Genomics. We chose sequencing platforms (HiSeq, NovaSeq) based on the total sequencing reads we need in each experiment (up to the number of samples in each multiplexing experiment). We have added more details to the methods to clarify these choices (Page 12, 13).

2) Figure legends and Supplemental Figure legends need to provide much more details. For example Fig S5, its not clear what we are looking at.

Response: We apologize to the reviewer for the unclear legends. We have gone through all legends to improve them by adding more details.

3) There are many figures and supplements showing UMAPs of unsupervised clustering of cell types, but there were v few images of feature plots of individual genes that were described in the manuscript. Some violin plots were shown but few individual gene feature plots.

Response: Thanks. We have added feature plots for some genes (Fig S3, S19).

4) The group could have done more bioinformatics analysis to generate cell fate trajectories for a subset of their data providing more depth on top of the one image of pseudotime vs real time (Figure 2A). How can this be easily interpreted to gain some new biological insights of genes that drive cell fate decisions?

Response: We thank the reviewer for the comment. We have identified the genes that expressed differentially along the pseudotime progressions in each cell type. We have plotted the top genes and provided them as a supplemental figure (Fig S15). Additionally, we would like to mention that we have also identified the gene modules, gene pathways, and transcription factors that express differentially along the stages in each cell type (Fig 2B, S14, S16).

5) They refer to Sup Fig. SA2, and Sup Fig SA3 in the Methods in the section, "Data alignment and cell type analysis". Which figure is that?

Response: We apologize for the confusion. We have corrected them. The correct citations should be Fig 1E and Fig S8A.

REVIEWER COMMENTS

Reviewer #1 (Remarks to the Author):

The authors have largely addressed my prior concerns. This data-rich manuscript will be a useful resource for the field. I have three remaining minor points:

1. Please say more in the results about the two genes differing between the strains analyzed that were selected for validation. At a minimum this should indicate their names and what is known about their function.
2. In the last line of the abstract, please remove "the" between "for" and "studies".
3. The authors should reconsider the title of the manuscript (in particular use of the term "principles") as the complete transcriptional network of murine heart development is not deduced. Moreover, part of the authors' analysis addresses intercellular signaling, albeit deduced from transcriptomic data.

Reviewer #2 (Remarks to the Author):

The authors addressed several of the points in the prior review. Overall this remains a descriptive survey of heart development using single cell RNA-seq. The data obtained will be useful, but a major weakness of this manuscript is the low number of cells (average about 400) per sample (which the authors describe as a weakness of previous scRNAseq data). The manuscript does not make notable conceptual or mechanistic advances based on the dataset.

Major:

1. The title "transcriptional principles..." should be changed since transcriptional principles were not identified.
2. Since the differences between CD1 and C57BL6 were small, why not pool the data to increase the number of cells per sample? Because of the low coverage, the authors should temper their statements in the manuscript and abstract describing this as a complete or comprehensive dataset.
3. Multi-seq. The authors use many supplemental figures to cover the use of multi-seq. One undesirable aspect of multi-seq is that it appears that up to 1/3 of cells are doublets or lack a barcode.
4. Mouse strain. The authors indicate that they chose C57BL6 and CD1 to examine differences between mouse strains. C57BL/6 is an inbred strain often used for cardiac studies. CD1, on the other hand, is an outbred strain that is not often used for cardiac studies. It remains unclear why this experimental design was chosen. C57BL/6 belongs to two different substrains with known genetic differences: C57BL/6J and C57BL/6N. The authors do not indicate which was used.
5. Tissue samples. There is no information about how samples from each chamber were collected. Were RA and LA actually the atrial appendages? How were LV and RV samples dissected – did they include the AV canal region, including the endocardial cushions and the AV canal myocardium? The ventricular septum? Perhaps ventricular samples were free wall samples and did not include septum – hence the lack of septum genes such as *Irx1* and *Irx2*?
6. Suppl Tables: These were very hard to decipher since they are delivered to the reviewer with arbitrary names and they do not contain a Table name and legend within the file. Also some of the tables contain difficult to interpret entries like "transfac_pro__M04758" which may not even be accessible to users without a transfac pro subscription.

7. Fig. 3E. Naming the clusters A1 and V1 for atrial and ventricle does not correspond to the data, which shows that A1 and V1 each have both atrial and ventricular genes. But perhaps there is a predominance of LA and LV cells in "A1" and RA and RV cells in "V1"? A stacked bar plot showing the distribution of chamber types in these clusters would help (perhaps in place of 3F). This does not match with the patterns observed for *Sfrp2* and *Mest*, however. What is the interpretation of the very different distribution of A1 and V1 on sequential days in 3F? This is unlikely to be real biological signal. *Dcn* is described as a pan-Fb gene, yet it appears as an endoEC gene differentially expressed between RA2 and LA2.

8. *BMP10* is silenced in ventricle by about E16.5. It is transiently expressed in trabecular myocardium from about E9-E13.5. Therefore the "trabecular myocardium" cluster identified by *BMP10* corresponds to fetal trabeculations but not later embryonic or neonatal trabecular myocardium. This suggests that these later trabecular cardiomyocytes are elsewhere among the ventricular CMs – and that the statement that all but clusters 4 and 9 are compact cardiomyocytes is not accurate.

9. Fig. 5E: There is not good overlap between *Cldn5* and *Fabp4*. Some cells are *Cldn5*⁺ *Fabp4*⁻, others are *Cldn5*⁻ *Fabp4*⁺, and some are positive for both.

10. Fig. 6. This section links CHD genes to expression patterns from the single cell data. The genes mentioned in this section have well described expression patterns, so little new information appeared here. Some information was inaccurate, due to limitations of scRNAseq data. For example, *Tbx5* is known to be expressed in endocardial cells, where it is functionally important for valve development and cardiac septation. However, the scRNAseq data describes *Tbx5* and cardiomyocyte specific.

11. Fig. 7. Biological experiments showed that epicardial cells secrete *Igf2* to stimulate ventricular CM growth. The ligand receptor analysis did not find this – rather the analysis suggests that epicardial cells are the target of *Igf2*, not its source. This is not accurately described in the text. In the section referring to new findings about epicardial stimulation of postnatal heart growth, the authors should be more specific about which signaling pathways are predicted. Is there any validation of a subset of these predictions?

12. Fig. 7D: This experiment requires ANOVA with adjustment for multiple testing. What is the justification for using a 1-tailed test?

13. Fig. 8. For the analysis of WT and mutant cells, the authors primarily used one sample at a single stage. The number of epicardial cells that was the focus of the analysis was low – Wt1: 74 WT, 50 Mut; Tbx18: 49WT, 149 Mut. Replicates with more cells per replicate would be desirable to increase power and demonstrate reproducibility.

14. In 8J, It appears *Vcan* was not differentially expressed in 8J, yet the text focuses on potential role of *Vcan* in inhibiting cell proliferation. The legend should specify the statistical test used.

Minor:

1. Please label the figures numbers and put the legends on the figures.

2. In fig 1, E9.5 is described as a 4-chambered heart. There are no septae yet, so this is not accurate. It could be described as a looped heart tube. The cartoons are of mature human hearts, not developing hearts. They could be replaced by actual whole mount heart images that the authors show in the suppl. Figures.

3. In the discussion, the authors cite a reference that LA fibroblasts arise from epicardium and RA fibroblasts come from neural crest. The citation is incorrect and does not cover this topic. What is the evidence supporting this statement?

Reviewer #3 (Remarks to the Author):

Review of Feng et al.,

This is an improved manuscript because it provides some more details and insights than the original version. Major improvements include 1) clarifying confusions raised by the reviewers; 2) including more details in the analyses and figure legends; 3) expansion of the cell proliferation studies; 4) in situ hybridization validation of many genes and immunofluorescence staining; 5) more clarity as to proving insights from this data to understand the cause of CHD; Tgfb3 functional studies in Figure 8I, J. These make the manuscript stronger. My comments could be addressed as minor revisions.

The manuscript is stated to be an atlas on heart development and it is therefore a good resource in the field. It doesn't quite address a burning hypothesis for cardiac development. Each section probably could be its own research program. It would have been a plus if there was a particular biological hypothesis that was addressed rather than testing many hypotheses at once and providing sets of genes for individual in depth study. Major criticisms that can be easily addressed by adding a line or two.

- 1) CD1 vs Bl/6 mice. The authors provide more data regarding the comparison of the hearts in these two different strains, but they still did not provide a rationale at looking at these strains to begin with. They should provide one sentence in the Results that provides a convincing and clear biological rationale as to why they wanted to investigate two strains, and perhaps, these two strains. The response to Reviewer 2 by the authors, provides some biological rationale for looking at these 2 strains, such as difference in penetrance of cardiac phenotypes in these different strains, although the authors didn't provide a reference, but it would help in the writing to have a biological rationale in addition to just a technical rationale that the data was available. Then this can be discussed later in the manuscript in the CHD section as well, linking different sections somehow.
- 2) For the statement in regard to analysis of TFs, with the conclusion that early stage versus late stage TFs are not shared. Is there a particular insight or take home from this? It is then stated that Mef2a is expressed in neonatal stage atrial CMs and is predicted to regulate neonatal genes. This paragraph ends and we never see Mef2a in the text again. There are many themes touched upon, including some quite well, but the different sections don't build up to create a simple model testing a particular hypothesis.
- 3) For the phenotype analysis of Wt1 and Tbx18 global null mutants, it's not clear what the phenotypes of the hearts are (no histology sections were provided; Fig S27) for us to know how these are the same or different than what was reported. Esp since the next to last paragraph of the Introduction brings up a controversy in the field. Besides having more cells at E14.5 for scRNA-seq, was there a biological rationale for choosing E14.5 for analysis of Wt1 and Tbx18 mutants? If so, it would be good to provide that in the first paragraph on page 9.
- 4) It should be mentioned as a limitation of the study that one of the reasons for less ventricular cells in Fig S11 is because they are not captured by Chromium 10x, only if nuclei were purified would you get more even representation.

Minor points:

- 1) Gene names need to be italicized and for mouse, first letter is capitalized; I noticed that *tbx18* was indicated as such in the text and figures, but also as Tbx18, but not italicized.
- 2) What does "outperformed their current lineage markers" mean in first paragraph on Page 3?

- 3) Page 5, should be Fig S14B, not S14B;
- 4) Page 5, bottom paragraph: Ddig4l or Ddit4l?
- 5) Page 7, CHD list-provide reference, as indicated in the response to reviewers, but the reference is missing (Jin et al.,).
- 6) Page 8 top paragraph, I don't see Nrg1 or Igf in Fig 7B
- 7) Fig 8A, font on Y axis is too small. Same for gene names in D and E.
- 8) Are the Tbx18 and Wt1 null mutants? This isn't clear in the writing of the section on epicardium function(page 8).
- 9) In the Discussion, its still not clear how the scRNA-seq data helps us understand the mechanism for CHDs.
- 10) The grammar in the yellow highlighted section in the second paragraph of the Discussion needs to be fixed.

REVIEWER COMMENTS

Reviewer #1 (Remarks to the Author):

The authors have largely addressed my prior concerns. This data-rich manuscript will be a useful resource for the field. I have three remaining minor points:

1. Please say more in the results about the two genes differing between the strains analyzed that were selected for validation. At a minimum this should indicate their names and what is known about their function.

Response: We thank the reviewer for the comment. We have added more details on the two genes that were differentially expressed between the strains (page 4).

2. In the last line of the abstract, please remove "the" between "for" and "studies".

Response: We appreciate the reviewer for the comment. We have updated the abstract as suggested.

3. The authors should reconsider the title of the manuscript (in particular use of the term "principles") as the complete transcriptional network of murine heart development is not deduced. Moreover, part of the authors' analysis addresses intercellular signaling, albeit deduced from transcriptomic data.

Response: We thank the reviewer for the suggestion. We have updated the title to "Identification of stage and chamber-specific molecular features in murine embryonic and neonatal hearts through single-cell transcriptomic analysis."

Reviewer #2 (Remarks to the Author):

The authors addressed several of the points in the prior review. Overall this remains a descriptive survey of heart development using single cell RNA-seq. The data obtained will be useful, but a major weakness of this manuscript is the low number of cells (average about 400) per sample (which the authors describe as a weakness of previous scRNAseq data). The manuscript does not make notable conceptual or mechanistic advances based on the dataset.

Response: We appreciate the reviewer for the summary. Please find our responses to each specific point below.

Major:

1. The title "transcriptional principles..." should be changed since transcriptional principles were not identified.

Response: We thank the reviewer for the comment. We have updated the title to "Identification of stage and chamber-specific molecular features in murine embryonic and neonatal hearts through single-cell transcriptomic analysis."

2. Since the differences between CD1 and C57BL6 were small, why not pool the data to increase the number of cells per sample? Because of the low coverage, the authors should temper their statements in the manuscript and abstract describing this as a complete or comprehensive dataset.

Response: We thank the reviewer for the comment. Although we did not observe genome-wide transcriptional differences between the single cells in the two strains, we did identify a group of genes that differentially expressed between them. To avoid the effects of strain differences on downstream

analysis, keeping the two datasets separated will work better. Additionally, keeping the strains separated will help other labs use the datasets and repeat the findings. Lastly, based on the reviewer's suggestion, we have updated the abstract and manuscript to avoid describing the datasets as complete or comprehensive.

3. Multi-seq. The authors use many supplemental figures to cover the use of multi-seq. One undesirable aspect of multi-seq is that it appears that up to 1/3 of cells are doublets or lack a barcode.

Response: We thank the reviewer for the comment. The MULTI-seq has the advantage of multiplexing samples, which can reduce the cost and batch effects, but it also led to some sequencing reads being wasted as some profiled cells were discarded for having multiple or no barcodes. However, overall, MULTI-seq has significantly reduced the cost per sample. We have added a discussion on this point in the manuscript (page 10-11).

4. Mouse strain. The authors indicate that they chose C57BL6 and CD1 to examine differences between mouse strains. C57BL/6 is an inbred strain often used for cardiac studies. CD1, on the other hand, is an outbred strain that is not often used for cardiac studies. It remains unclear why this experimental design was chosen. C57BL/6 belongs to two different substrains with known genetic differences: C57BL/6J and C57BL/6N. The authors do not indicate which was used.

Response: We respectively disagree with the reviewer on the comment. CD1 mice have also been often used in cardiac studies (PMID: 12149465; PMID: 18722982; PMID: 8272575). However, because of the relatively low penetrance in developing cardiac phenotypes, they were mainly used to study normal heart developmental processes. In this study, we have generated single cell datasets for both strains, which will be valuable resources for studying heart development under normal and diseased conditions. Additionally, the comparative analysis of CD1 and C57BL/6 mouse hearts has been performed before and strain-associated differences like cardiac repolarization and response to acute hypoxia have been identified (PMID: 15621042; PMID: 16916919). We have described more on the rationale of profiling the two strains in the manuscript (page 3). Lastly, we used C57BL/6N in the study, which we ordered from the Charles River Laboratory. We have added the information in the methods section (page 12).

5. Tissue samples. There is no information about how samples from each chamber were collected. Were RA and LA actually the atrial appendages? How were LV and RV samples dissected – did they include the AV canal region, including the endocardial cushions and the av canal myocardium? The ventricular septum? Perhaps ventricular samples were free wall samples and did not include septum – hence the lack of septum genes such as *Irx1* and *Irx2*?

Response: We thank the reviewer for the comments and apologize for not providing the information. As we have described previously (PMID: 27840109; PMID: 30283141), we dissected the hearts into four chambers based on anatomical landmarks such as the septal groove between the LV and RV and between the LA and RA. The LA and RA we isolated are atrial chambers, visible in Fig S4. Regarding the AV canal and ventricular septum, we collected them together with the LV samples. As shown in Fig S19B-D, the AVC cells (cluster 9) and septum cells (*Irx1* and *Irx2* positive cells) were mainly from the LV samples. We have added these details to the materials and methods section (page 12).

6. Suppl Tables: These were very hard to decipher since are delivered to the reviewer with arbitrary names and they do not contain a Table name and legend within the file. Also some of the tables contain

difficult to interpret entries like “transfac_pro__M04758” which may not even be accessible to users without a transfac pro subscription.

Response: We apologize for the inconvenience. We provided the table legends in a separate file in our last submission. This time, we have added the table name and legend into each table file and hope this will be more convenient to read. Regarding entries like “transfac_pro__M04758”, they represent the motifs that RcisTarget analysis identified on the genomic elements of target genes. As they are not essential to understanding the prediction results, we have decided to remove them in Table S5 and S12 to avoid confusion.

7. Fig. 3E. Naming the clusters A1 and V1 for atrial and ventricle does not correspond to the data, which shows that A1 and V1 each have both atrial and ventricular genes. But perhaps there is a predominance of LA and LV cells in “A1” and RA and RV cells in “V1”? At stacked bar plot showing the distribution of chamber types in these clusters would help (perhaps in place of 3F). This does not match with the patterns observed for Sfrp2 and Mest, however. What is the interpretation of the very different distribution of A1 and V1 on sequential days in 3F? This is unlikely to be real biological signal. Dcn is described as a pan-Fb gene, yet it appears as an endoEC gene differentially expressed between RA2 and LA2.

Response: We apologize to the reviewer for the confusion, which was caused by a mistake we made. We incorrectly labeled the zones in the revised Fig 3E (RA and LV were swapped). We have updated the figure with the correct labels and believe all the related questions in the comment have been solved.

Regarding the Dcn expression, according to our single cell RNA-seq data, Dcn is highly expressed in fibroblast, but it also had expression in the epicardial cell and Endo_EC (Letter Fig1). We also analyzed the expression of the other fibroblast genes Col1a1 and Postn. We found Col1a1 was expressed in fibroblast, epicardial cell, and mural cell, and Postn was expressed in fibroblast, Endo_EC, and mural cell (Letter Fig1). The results suggested that the fibroblast marker genes are generally not cell-type specific. To be accurate, we have updated the manuscript by replacing the “pan-Fb gene” with the “Fb-highly expressed gene.”

Letter Fig 1: Violin plot showing the expression pattern of Fb genes.

8. BMP10 is silenced in ventricle by about E16.5. It is transiently expressed in trabecular myocardium from about E9-E13.5. Therefore the “trabecular myocardium” cluster identified by BMP10 corresponds to fetal trabeculations but not later embryonic or neonatal trabecular myocardium. This suggests that these later trabecular cardiomyocytes are elsewhere among the ventricular CMs – and that the statement that all but clusters 4 and 9 are compact cardiomyocytes is not accurate.

Response: We thank the reviewer for the comment. As we have labeled compact cardiomyocytes based on the expression of multiple genes (Mycn and Hey2 positive, Bmp10 and Slit2 negative), we believe that most cells in the clusters except 4 and 9 are compact cardiomyocytes. However, we agree with the reviewer that some cells in these clusters maybe trabecular cardiomyocytes at the later stages due to the lack of specific marker genes. We have updated the description on page 6 to make it accurate.

9. Fig. 5E: There is not good overlap between Cldn5 and Fabp4. Some cells are Cldn5+ Fabp4-, others are Cldn5- Fabp4+, and some are positive for both.

Response: Thanks! We believe this happens because the two genes are expressed in partially different Vas_EC populations. Through plotting the two genes' expression on the same plots, we identified many Vas_EC co-expressing the two genes but also found some cells expressing only one of the two genes (Letter Fig2). We have updated the relevant descriptions in the manuscript to reflect this (page 7).

Letter Fig 2: Comparison of Fabp4 and Cldn5 expression patterns.

10. Fig. 6. This section links CHD genes to expression patterns from the single cell data. The genes mentioned in this section have well described expression patterns, so little new information appeared here. Some information was inaccurate, due to limitations of scRNAseq data. For example, Tbx5 is known to be expressed in endocardial cells, where it is functionally important for valve development and cardiac septation. However, the scRNAseq data describes Tbx5 and cardiomyocyte specific.

Response: We respectively disagree with the reviewer that all genes in this section have well-described expression patterns. Many of the CHD genes, in particular the ones in non-CMs, have never been analyzed at the single cell level across multiple developmental stages. Understanding the expression pattern (cell type and stage) of these genes at the single cell level is basic but critical to understanding their mechanisms in causing CHDs. Regarding the expression pattern of Tbx5, although we did observe its expression in Endo_EC, it was at a much lower level than in the CMs (Letter Figure 3). We have updated the manuscript by changing “cardiomyocyte specific” to “cardiomyocyte highly expressed” (page 7).

Letter Fig 3: The expression enrichment heatmap of CM highly expressed genes.

11. Fig. 7. Biological experiments showed that epicardial cells secrete Igf2 to stimulate ventricular CM growth. The ligand receptor analysis did not find this – rather the analysis suggests that epicardial cells are the target of Igf2, not its source. This is not accurately described in the text. In the section referring to new findings about epicardial stimulation of postnatal heart growth, the authors should be more specific about which signaling pathways are predicted. Is there any validation of a subset of these predictions?

Response: We thank the reviewer for the comments. We predicted the ligand-receptor interactions in each pathway using the software CellChat, which employed an unsupervised pattern recognition method to identify the global communication patterns between cell types.

(<https://htmlpreview.github.io/?https://github.com/sqjin/CellChat/blob/master/tutorial/CellChat-vignette.html>). In our original analysis, we used the default threshold value of 0.5 to uncover the most enriched interactions. However, when we changed the threshold to a less strict value of 1, we identified more signaling

pathway interactions associated with each cell type, including the interaction that the reviewer mentioned. Specifically, it's the interaction between epicardial cell derived Igf2 and Ven_CM derived Igf2r, which we have highlighted in Letter Fig 4. To keep the threshold value consistent in the analysis of each pathway, we have kept the results with original threshold value of 0.5 in the supplemental figure. We apologize for not describing the results accurately and have updated the related sentences in the manuscript (page 8).

Letter Fig 4: The predicted IGF pathway ligand-receptor interactions between different cell types.

Furthermore, we have added details in the manuscript on the predicted signaling pathways at postnatal stage (page 8). Further validation of these pathways is out of the scope of this study. However, importantly, we have validated the function of epicardial cell derived signals that were predicted to regulate the fetal to neonatal transitions (Fig 7D).

12. Fig. 7D: This experiment requires ANOVA with adjustment for multiple testing. What is the justification for using a 1-tailed test?

Response: As we were mainly interested in the comparisons between the controls and each growth factor treatment, we believe the student T-tests were appropriate in this analysis. We have improved the figure by labeling the comparison pairs to clarify it.

13. Fig. 8. For the analysis of WT and mutant cells, the authors primarily used one sample at a single stage. The number of epicardial cells that was the focus of the analysis was low – Wt1: 74 WT, 50 Mut; Tbx18: 49WT, 149 Mut. Replicates with more cells per replicate would be desirable to increase power and demonstrate reproducibility.

Response: As we were interested in how the defects in epicardial cells affect the development of other cardiac lineages in the two mutants (Wt1 and Tbx18), we aimed to profile all the main cardiac lineages. Since epicardial cell is a single cell layer of the heart, its proportion in the recovered cardiac cell types is relatively low. This will be the same when more replicates are carried out. Additionally, although the cell numbers are relatively small, we found the genes differentially expressed in the controls and mutants to be largely consistent across the cells (Fig 8Ci and ii, and S29D) and don't think the results will change after profiling more cells. Furthermore, through comparative analysis of the genes that are differentially

expressed in the Tbx18 mutant and the control epicardial cells at E14.5 and E17.5, we identified many genes and pathways in common at the two stages (Table S17, Letter Fig 5), suggesting that the

Pathway_Upregulated_GenePathways	Pathway_Downregulated_GenePathways
establishment of endothelial blood-brain barrier	regulation of mesenchymal cell proliferation
collagen biosynthetic process	cardiac septum morphogenesis
regulation of ventricular cardiac muscle cell action potential	outflow tract morphogenesis
detection of mechanical stimulus	positive regulation of cell adhesion
regulation of cell adhesion	neurogenesis
regulation of Wnt signaling pathway	muscle structure development

Letter Fig 5: The pathway of genes that differentially expressed in control and Tbx18 mutant epicardial cells at E14.5 and E17.5.

findings can be repeated. Lastly, we have provided the shared genes as a supplemental table (Table S17).

14. In 8J, It appears Vcan was not differentially expressed in 8J, yet the text focuses on potential role of Vcan in inhibiting cell proliferation. The legend should specify the statistical test used.

Response: Thanks. Vcan was differentially expressed between control and TGFB3 treated sample with a p value of 0.055392. We have added the p value to the plots and specified the statistical tests in the legend.

Minor:

1. Please label the figures numbers and put the legends on the figures.

Response: Thanks. We have labeled the figure numbers and put the legends on the figures.

2. In fig 1, E9.5 is described as a 4-chambered heart. There are no septae yet, so this is not accurate. It could be described as a looped heart tube. The cartoons are of mature human hearts, not developing hearts. They could be replaced by actual whole mount heart images that the authors show in the suppl. Figures.

Response: Thanks. We have updated the diagram by improving the labels and using real mouse embryonic hearts.

3. In the discussion, the authors cite a reference that LA fibroblasts arise from epicardium and RA fibroblasts come from neural crest. The citation is incorrect and does not cover this topic. What is the evidence supporting this statement?

Response: We apologize for the mistake. We have added the right reference (PMID: 26748307).

Reviewer #3 (Remarks to the Author):

Review of Feng et al.,

This is an improved manuscript because it provides some more details and insights than the original version. Major improvements include 1) clarifying confusions raised by the reviewers; 2) including more details in the analyses and figure legends; 3) expansion of the cell proliferation studies; 4) in situ hybridization validation of many genes and immunofluorescence staining; 5) more clarity as to proving insights from this data to understand the cause of CHD; Tgfb3 functional studies in Figure 8I, J. These make the manuscript stronger. My comments could be addressed as minor revisions.

The manuscript is stated to be an atlas on heart development and it is therefore a good resource in the field. It doesn't quite address a burning hypothesis for cardiac development. Each section probably could be its own research program. It would have been a plus if there was a particular biological hypothesis that was addressed rather than testing many hypotheses at once and providing sets of genes for individual in depth study. Major criticisms that can be easily addressed by adding a line or two.

Response: We appreciate the reviewer for the summary. As the reviewer specified, we have added a line or two in the manuscript to address each major criticism. Please find our responses to each specific point below.

1) CD1 vs Bl/6 mice. The authors provide more data regarding the comparison of the hearts in these two different strains, but they still did not provide a rationale at looking at these strains to begin with. They should provide one sentence in the Results that provides a convincing and clear biological rationale as to why they wanted to investigate two strains, and perhaps, these two strains. The response to Reviewer 2 by the authors, provides some biological rationale for looking at these 2 strains, such as difference in penetrance of cardiac phenotypes in these different strains, although the authors didn't provide a reference, but it would help in the writing to have a biological rationale in addition to just a technical rationale that the data was available. Then this can be discussed later in the manuscript in the CHD section as well, linking different sections somehow.

Response: We thank the reviewer for the suggestion. We have described the biological rationale of using both CD1 and C57BL/6 strains in the Results section (page 3). We have also discussed this issue in the CHD-related paragraph (page 11).

2) For the statement in regard to analysis of TFs, with the conclusion that early stage versus late stage TFs are not shared. Is there a particular insight or take home from this? It is then stated that Mef2a is expressed in neonatal stage atrial CMs and is predicted to regulate neonatal genes. This paragraph ends and we never see Mef2a in the text again. There are many themes touched upon, including some quite well, but the different sections don't build up to create a simple model testing a particular hypothesis.

Response: We thank the reviewer for the comments. Regarding the TFs analysis, we think the take-home message is that the cardiac cells at early embryonic and neonatal stages, but not late embryonic stages expressed unique TFs, which is consistent with their function in specifying cardiac lineages at early embryonic stage and adapting new environment at neonatal stage. Regarding Mef2a, we have added a sentence to describe its function and mutant phenotype (page 5). Furthermore, considering that our goal in this study is to generate an atlas of CMs and non-CMs in developing mouse hearts at embryonic and neonatal stages, we apologize to the reviewer for the difficulty of building up different sections in the manuscript to test a particular hypothesis.

3) For the phenotype analysis of Wt1 and Tbx18 global null mutants, it's not clear what the phenotypes of the hearts are (no histology sections were provided; Fig S27) for us to know how these are the same or different than what was reported. Esp since the next to last paragraph of the Introduction brings up a controversy in the field. Besides having more cells at E14.5 for scRNA-seq, was there a biological rationale for choosing E14.5 for analysis of Wt1 and Tbx18 mutants? If so, it would be good to provide that in the first paragraph on page 9.

Response: We thank the reviewer for the comments. We found that the Wt1 mutants at E13.5 and E14.5 had clear body wall edema, and their hearts had more rounded and bifid apices, consistent with what was reported previously (PMID: 21663736). In Tbx18 mutants, we did not observe obvious defects based on the heart morphology, which was also consistent with the previous reports (PMID: 24016759; PMID: 22926762). Although the previous reports had controversy, both studies did not identify obvious defects based on the heart morphology. One of the studies found vasculature defects through staining analysis with the SM22 α -lacZ reporter line; consistently, we also identified molecular deficiencies in Vas_EC using scRNA-seq. We have added a sentence describing the phenotype in both mutants (page 9).

Regarding the use of E14.5 for the analysis, besides this stage with the most abundant cells in the scRNA-seq data, we'd like to also emphasize that this stage is an ideal timepoint to compare the Vas_EC's in control and mutants since the Vas_EC's start to develop at E12.5 and becomes one of the main cardiac cell types at E14.5. We have also included this information in the manuscript on page 9.

4) It should be mentioned as a limitation of the study that one of the reasons for less ventricular cells in Fig S11 is because they are not captured by Chromium 10x, only if nuclei were purified would you get more even representation.

Response: We appreciate the reviewer for the suggestion. We have added a discussion of the limitations in page 11.

Minor points:

1) Gene names need to be italicized and for mouse, first letter is capitalized; I noticed that *tbx18* was indicated as such in the text and figures, but also as *Tbx18*, but not italicized.

Response: Thanks. We have corrected the gene names.

2) What does "outperformed their current lineage markers" mean in first paragraph on Page 3?

Response: We apologize for the confusion. We meant that the lineage genes we identified have better specificity than the current lineage markers. We have updated the description to make it clear.

3) Page 5, should be Fig S14B, not S14B;

Response: Thanks. We have updated it.

4) Page 5, bottom paragraph: *Ddig4l* or *Ddit4l*?

Response: Thanks. They should be *Ddit4l*. We have corrected them.

5) Page 7, CHD list-provide reference, as indicated in the response to reviewers, but the reference is missing (Jin et al.,).

Response: Thanks. We have added the reference.

6) Page 8 top paragraph, I don't see *Nrg1* or *Igf* in Fig 7B

Response: Thanks. Fig 7B has only stage-unique interactions. *Nrg1* and *Igf* and their receptors express at more than one stage, so they were not listed in that figure. We have analyzed these pathways (Fig S25) as they were reported before in heart development.

7) Fig 8A, font on Y axis is too small. Same for gene names in D and E.

Response: We thank the reviewer for the comment. We have improved the fonts in Fig 8A, D, and E.

8) Are the *Tbx18* and *Wt1* null mutants? This isn't clear in the writing of the section on epicardium function (page 8).

Response: Yes, they are null mutants. We have added the information on page 9.

9) In the Discussion, it's still not clear how the scRNA-seq data helps us understand the mechanism for CHDs.

Response: Thanks. We think the expression pattern (cell type and stage) of the CHD genes is basic but critical information to understanding their mechanisms in causing the CHDs. More work will surely be required to elucidate the function of these genes. We have updated the sentence in Discussion (page 11) to improve the description.

10) The grammar in the yellow highlighted section in the second paragraph of the Discussion needs to be fixed.

Response: Thanks. We have fixed the grammar in the second paragraph of the Discussion.

REVIEWER COMMENTS

Reviewer #2 (Remarks to the Author):

The authors responded productively to most of my prior comments.

Remaining minor comments:

Fig. 7D. This should be ANOVA with Dunnett's posthoc test to compare each condition to control. It should be two-tailed and not one-tailed unless the authors can justify a one-tailed test.

Fig 8J. $P=0.055$ is above the threshold for statistical significance, $P<0.05$. Here again the authors should use a two-tailed test unless good justification is provided for a one-tailed test.

Regarding the sentence in the discussion about the origin of cardiac fibroblasts: "Although there are no reports on the different developmental sources for the CMs in LA and RA, the fibroblasts in LA and RA are known to be derived from different sources, with LA fibroblasts developing from epicardium and RA fibroblasts coming from the neural crest^{67,68}", this statement does not accurately reflect the referenced review or the original manuscript. These point out that rare neural crest-derived fibroblasts can be observed in RA, but there is not statement that the predominant source of fibroblasts in RA is from neural crest.

"We did not identify obvious heart defects in the mutants based on their morphology". Wt1 and Tbx18 mutant hearts have been well described to have morphological abnormalities. The gross heart images shown in Fig. S27 do show subtle morphological differences in mutants, and the differences would have been more pronounced if histological sectioning was performed. The authors also describe abnormalities in the Wt1 mutant hearts just a few sentences earlier. Therefore the authors should modify this statement in the text.

Reviewer #3 (Remarks to the Author):

The manuscript is improved; there is still one dataset that would be beneficial to readers, and it would be to provide a histopathological analysis of the hearts in Tbx18 and Wt1 null mutant embryos. In the response to the review, the authors provided some explanation of gross phenotypes. Some of this is provided in the revised text, and some isn't; this can be easily added to the text. Histological analyses of tissue sections of the hearts would be greatly helpful of Tbx18 and Wt1 null mutants, in their hands, especially, since there is a whole figure, Fig 8, provided in the manuscript, based upon analyses of mutants versus controls.

Minor: Please italicize gene and mRNA names. Starting with, e.g.: Gene/mRNA names *Sln* and *Myl2* should be italicized in "After quality control, filtering and normalization (Fig S2B-F), expression analyses of the atrial CM gene *Sln* and ventricular CM gene *Myl2* showed that *Sln*+ CMs were assigned to atrial samples (LA and RA), and *Myl2*+ CMs were assigned to ventricular samples (LV and RV) (Fig S3A, B)." And, figures, e.g. Figure 1E, gene names should be italicized. This is especially noted in the new additions to the current revision. This occurs randomly throughout the manuscript text and figures.

REVIEWER COMMENTS

Reviewer #2 (Remarks to the Author):

The authors responded productively to most of my prior comments.

Remaining minor comments:

Fig. 7D. This should be ANOVA with Dunnett's posthoc test to compare each condition to control. It should be two-tailed and not one-tailed unless the authors can justify a one-tailed test.

Response: We appreciate the reviewer for the comment. As the reviewer suggested, we have compared each condition to the control using ANOVA with Dunnett's posthoc test. We have updated figure 7D and the related descriptions in the manuscript with the new testing results (page 8).

Fig 8J. $P=0.055$ is above the threshold for statistical significance, $P<0.05$. Here again the authors should use a two-tailed test unless good justification is provided for a one-tailed test.

Response: We thank the reviewer for the comment. We have recalculated the statistical significance by using two-tailed tests. The results remain the same, with the P values in Fbn2 and Gpc3 expression quantifications being below 0.05. Considering Vcan is not statistically significant, we have removed the description of it in the manuscript (page 10).

Regarding the sentence in the discussion about the origin of cardiac fibroblasts: "Although there are no reports on the different developmental sources for the CMs in LA and RA, the fibroblasts in LA and RA are known to be derived from different sources, with LA fibroblasts developing from epicardium and RA fibroblasts coming from the neural crest^{67,68}", this statement does not accurately reflect the referenced review or the original manuscript. These point out that rare neural crest-derived fibroblasts can be observed in RA, but there is not statement that the predominant source of fibroblasts in RA is from neural crest.

Response: We appreciate the reviewer for the comment and apologize for not citing the statement accurately. We incorrectly interpreted one of the diagrams in the referenced review paper. After carefully reading the review and original manuscripts, we agree with the reviewer that only part of the fibroblasts in RA was derived from the neural crest. To avoid confusion, we have deleted the related sentence in the manuscript (page 11).

"We did not identify obvious heart defects in the mutants based on their morphology". Wt1 and Tbx18 mutant hearts have been well described to have morphological abnormalities. The gross heart images shown in Fig. S27 do show subtle morphological differences in mutants, and the differences would have been more pronounced if histological sectioning was performed. The authors also describe abnormalities in the Wt1 mutant hearts just a few sentences earlier. Therefore the authors should modify this statement in the text.

Response: We apologize to the reviewer for the confusion. In that sentence, we meant that we did not identify obvious heart defects in the Tbx18 mutants based on their morphology. We have updated the sentence to be accurate (page 9). Additionally, we have performed histological section and

immunofluorescence analysis in *Wt1* and *Tbx18* mutant hearts. We observed thinner ventricular myocardium in the *Wt1* mutant and ectopic nodules with CD31-positive cells in the *Tbx18* mutant hearts. These observations are consistent with what was reported previously (Wu et al., *Developmental Biology*, 2013). We have included the results in two supplemental figures (Fig S28, 29) and described them in the manuscript (page 9).

Reviewer #3 (Remarks to the Author):

The manuscript is improved; there is still one dataset that would be beneficial to readers, and it would be to provide a histopathological analysis of the hearts in *Tbx18* and *Wt1* null mutant embryos. In the response to the review, the authors provided some explanation of gross phenotypes. Some of this is provided in the revised text, and some isn't; this can be easily added to the text. Histological analyses of tissue sections of the hearts would be greatly helpful of *Tbx18* and *Wt1* null mutants, in their hands, especially, since there is a whole figure, Fig 8, provided in the manuscript, based upon analyses of mutants versus controls.

Response: We thank the reviewer for the comment. We have performed histological section and immunofluorescence analysis of CD31 and cTNT in *Wt1* and *Tbx18* mutant hearts. We observed thinner ventricular myocardium in the *Wt1* mutant and ectopic nodules with CD31-positive cells in the *Tbx18* mutant hearts. These observations are consistent with what was reported previously (Wu et al., *Developmental Biology*, 2013). We have included the results in two supplemental figures (Fig S28, 29) and described them in the manuscript (page 9).

Minor: Please italicize gene and mRNA names. Starting with, e.g.: Gene/mRNA names *Sln* and *Myl2* should be italicized in “After quality control, filtering and normalization (Fig S2B-F), expression analyses of the atrial CM gene *Sln* and ventricular CM gene *Myl2* showed that *Sln*+ CMs were assigned to atrial samples (LA and RA), and *Myl2*+ CMs were assigned to ventricular samples (LV and RV) (Fig S3A, B).” And, figures, e.g. Figure 1E, gene names should be italicized. This is especially noted in the new additions to the current revision. This occurs randomly throughout the manuscript text and figures.

Response: We thank the reviewer for the comment. We have gone through the manuscript and figures to italicize the gene names.